# Simultaneous assimilation of water levels from river gauges and satellite flood maps for near-real time flood mapping

Antonio Annis[1,2], Fernando Nardi[1,3], and Fabio Castelli[2]

[1]WARREDOC, University for Foreigners of Perugia, Perugia Italy
[2]DICEA, University of Florence, Florence, Italy
[3]Institute of Water and Environment, Florida International University, Miami, USA
**Correspondence:** Antonio Annis (antonio.annis@unistrapg.it)

**Abstract.** Hydro-meteo hazard Early Warning Systems (EWSs) are operating in many regions of the world to mitigate nuisance effects of floods. EWSs performances are majorly impacted by the computational burden and complexity affecting flood prediction tools, especially for ungauged catchments that lack adequate river flow gauging stations. Earth Observation (EO) systems may surrogate to the lack of fluvial monitoring systems supporting the setting up of affordable EWSs. But, EO data, constrained by spatial and temporal resolution limitations, are not sufficient alone, especially at medium-small scales. Multiple sources of distributed flood observations need to be used for managing uncertainties of flood models, but this is not a trivial task for EWSs. In this work, a near real-time flood modelling approach is developed and tested for the simultaneous assimilation of both water level observations and EO-derived flood extents. An integrated physically-based flood wave generation and propagation modelling approach, that implements a Ensemble Kalman Filter, a parsimonious geomorphic rainfall-runoff algorithm (WFIUH) and a Quasi-2D hydraulic algorithm, is proposed. An approach for assimilating multiple stage gauges observations is proposed to overcome stability issues related to the updating of the Quasi-2D hydraulic model states. Furthermore a methodology to retrieve distributed observed water depths from satellite images to update 2D hydraulic modelling state variables is implemented. Performances of the proposed approach are tested on a flood event for the Tiber river basin in central Italy. The selected case study shows varying performances depending if local and distributed observations are separately or simultaneously assimilated. Results suggest that the injection of multiple data sources into a flexible data assimilation framework, constitute an effective and viable advancement for flood mitigation tackling EWSs uncertainty and numerical stability issues. Specifically, our findings reveal that the simultaneous assimilation of observations from static sensors and satellite images led to an overall improvement of the NSE between 5 and 40%, the Pearson correlation up to 12% and Bias reduction up to 80% as respect to the Open Loop simulation. Moreover, this combined assimilation allows to reduce the flood extent uncertainty as respect to the disjoint assimilation simulations for several hours after the satellite image acquisition.

## 1 Introduction

Floods represent one of the most costly and deadly natural disasters (EM-DAT, 2016), affecting annually on average more than 21 million people and producing economic loss greater than US $ 100 billion (Desai et al., 2015). The ability to understand and

predict floods represent a crucial assets of river basin management strategies (Knight and Shamseldin, 2005). Numerical simulations of flood scenarios are used for proper design of structural (e.g. levees, diversion channels, dams, etc.) and non-structural mitigation measures (e.g. land use regulations, flood zoning, flood proofing, flood forecasting and warning, disaster prevention, preparedness and response, Thampapillai and Musgrave, 1985). Early Warning Systems (EWSs) are nowadays increasingly used for timely detection of flood events (Kundzewicz, 2013).

EWSs generally require integrated geospatial modelling of floodplain domains supporting integrated topographic-hydrologic-hydraulic modelling chains to produce inundation predictions (Krzhizhanovskaya et al., 2011). Digital terrain models (DTMs), rainfall and runoff observations are required by EWSs for flood nowcasting and forecasting (Grimaldi et al., 2016). In case of medium term forecast (i.e. days/weeks ahead), rainfall and runoff observations are not sufficient and Numerical Weather Prediction (NWP) models are required, especially for basins whose concentration time is limited so that emergency measures, such as evacuation, cannot be properly applied on time (Hopson and Webster, 2010). In this regard, recent advances in NWP models in weather forecasting were developed adopting Ensemble Prediction Systems (EPS) (Buizza et al., 2005) as inputs of hydrological and hydraulic models. Flood models are computer- and data-intensive applications with input data requirements (i.e. accuracy and distribution) that are often unmet, especially river flow observations (Wing et al., 2020). As a result, EWSs suffer of several concurring uncertainties associated to boundary conditions, numerical parameterizations and discretizations of floodplain features and processes (Demeritt et al., 2007). The calibration and validation of flood models for data-scarce regions constitute, thus, a significant challenge for flood modellers that are often compelled to understand and manage parameter and output uncertainties (Moradkhani et al., 2005; Hostache et al., 2011; Liu and Gupta, 2007).

Data Assimilation (DA) methods represent effective means to reduce these uncertainties (Cloke and Pappenberger, 2009). DA improve EWS performances by adjusting flood model parameters, input, output or state variables using available observations. DA models are used both in NWP and hydrologic-hydraulic modelling.

Advances in EPS approaches and increasing of computational power allowed to improve the accuracy of NWP models as inputs of flood forecasting systems (Yu et al., 2016). Successful examples of advanced EPS approaches in NWP models for flood forecasting services at large scale are the EPS-ECMWF - from the European Centre for Medium Range Weather Forecasts- (De Roo et al., 2003) and the COSMO-LEPS - from the Consortium for Small-Scale Modelling – Limited-area Ensemble Prediction System (Marsigli et al., 2005).

Flood models can be updated in DA approaches by ingesting outputs of NWP models or direct rainfall-runoff observations. Stream gauge observations are the most used for updating hydrologic (McLaughlin, 2002; Moradkhani et al., 2005; Liu and Gupta, 2007) and hydraulic (Madsen and Skotner, 2005; Neal et al., 2007) model variables. However, single (or sparse) gauging stations generally fail to provide accurate flow observations during extreme events due to the distributed complex nature of flood processes (e.g. split flows, tributary junctions, overbank flow conditions, bridge overtopping). This is particularly critical for EWSs covering secondary ungauged river networks (Biancamaria et al., 2011; Mason et al., 2012). Moreover, we are observing a global decrease of the gauging stations in the river network (Stokstad, 1999; Sivapalan et al., 2003; Schumann et al., 2015).

To tackle these issues, in the last ten years, Earth Observation (EO) data were used to inject water altimetry observations in DA frameworks for updating flood models, usually adopting radar Synthetic Aperture Radar (SAR) technologies and 1D ( Matgen et al., 2007; Neal et al., 2009; Matgen et al., 2010; Giustarini et al., 2011) or 2D ( Andreadis et al., 2007; Hostache et al., 2010; Mason et al., 2012; García-Pintado et al., 2013; Andreadis and Schumann, 2014) hydraulic routing algorithms. One of the critical issues of the model state updating is the persistence of the improvements of the model performances. Regardless the DA algorithm (e.g. Direct Insertion, PF, EnKF) the assimilation of the model states in real and synthetic scenarios brought to more accurate predictions immediately after the updating step, and they quickly decrease, depending on the specific case study, few hours or even few minutes after the state updating, going back to the same performances of the open-loop model realisation (Andreadis et al., 2007; Matgen et al., 2010; García-Pintado et al., 2013; Andreadis and Schumann, 2014). Some of these studies demonstrated that the updating of inflows boundaries can increase the persistence of the errors reductions between the observations in both 1D (Matgen et al., 2010) and 2D (Andreadis et al., 2007; García-Pintado et al., 2013) hydraulic models. Other studies investigated on the spatial weighting of remote sensing-derived water levels observations in DA approaches (Grimaldi et al., 2016). For example Giustarini et al. (2011) found significant benefits in a local weighting procedure of assimilating unbiased very precise water levels observations, while a global weighting procedure is recommended for water level observations in ungauged basins. However, if the local weighting is combined with poorly spatially distributed field data, the model updating can lead to an over-correction that could even decrease the overall model performances. In fact, the frequency of the model corrections seems to be effective mostly during the rising limb of the flow hydrograph, while it seems not to be significant efficient during the recession limb (Giustarini et al., 2011; García-Pintado et al., 2013). García-Pintado et al. (2015) proposed a novel methodology to test the performance of a global formulation, a traditional local formulation and their own novel local formulation of the EnKF model to improve the forecast of a 2D hydraulic model assimilating SAR derived water levels. Their novel local formulation of the EnKF was able to remove the unphysical relationships and spurious correlations that characterized the global filter. The authos also proved that the updating of the 2D hydraulic model friction and channel bathymetry seems to have second order effect, as respect to the inflow updating, in flood inundation models applied to gradually varied flow in large rivers. Andreadis and Schumann (2014) applied a local EnKF for assimilating synthetic SAR derived water levels, inundation width and flood extent in a 2D hydraulic model, partitioning the Ohio River (516 km) in reaches of equal lengths. The authors obtained similar results for reach lengths varying from 5 to 50 km.

Beside satellite derived altimetry, in the last years SAR-derived inundation extent mapping techniques were tested to provide spatially distributed information to support near real-time flood detection services (Martinis et al., 2015; Pierdicca et al., 2009).

There are recent examples of DA research proving the value of assimilating satellite images for diverse purposes. In this regard, several aspects have been investigated to assimilate flood extents observations in a flood forecasting framework, such as the relation between the flood extent and the model state variable, the updating of the model inflows and parameters, the impact of the typology, the timing, the location and the frequency of the satellite-derived flood extent observations on the performances of the DA performances. Lai et al. (2014) applied a variational data assimilation (4D-Var) method for updating the friction (i.e. Manning the values) of a 2D hydraulic model based on shallow water equations proposing a novel cost function able to relate the satellite derived flood extent to indirectly observed flood depths. Revilla-Romero et al. (2016) proposed an EnKF approach

for updating the streamflow values and the parameters of a Global rainfall-runoff (LISFLOOD) model using flood extent observations gathered from the Global Flood Detection System (GFDS). Hostache et al. (2018) proposed a PF approach for updating an high resolution hydraulic model directly using ENVISAT ASAR derived water extents for near-real time flood forecasting. The authors analyzed improved performances of EWSs in reducing water level estimation errors if compared to Open Loop (OL) simulations (i.e. not updating flood state variables with observations). Hostache et al. (2018) underlined opportunities of SAR images, overcoming visibility issues of optical sensors due to clouds, but also stressing some limitations of water altimetry approaches. In particular, the need of high resolution topographic data, challenging pre-processing and hydraulic modelling development make SAR-derived DA approaches hard to replicate and to be applied at varying scales (Mason et al., 2012; Wood et al., 2016). Dasgupta et al. (2021b) proposed a novel Mutual Information-based likelihood function for assimilating SAR derived flood extents in an high resolution 2D hydraulic model adopting a PF approach. Dasgupta et al. (2021a) investigated on the timing, the positioning and the frequency of the SAR-derived flood extents, on the performances of the PF assimilation of a 2D hydraulic model, finding that the optimal strategy for the image acquisition depends on the river morphology and flood wave arrival timing. Moreover it was found that the number of observations to significantly improve the performances of the DA model increase with the with the narrowing of the floodplain valley.

Despite the remarkable progress in the integration of remotely sensed observations in DA frameworks, there are still major challenges (Grimaldi et al., 2016). For example, an approach able to assimilate heterogeneous observations from both local and distributed datasets coming from different sources (i.e. traditional stage gauges and remotely sensed flood extents) is still missing. Moreover, Quasi-2D and 2D hydraulic models can be sensitive to different simultaneous local state updating (i.e. water level corrections at specific time steps), because contiguous channel/floodplain cells can be characterized by different elevations, geometry and roughness, therefore instability issues can rise during the model state corrections with standard localization techniques. Another critical issue is that large scale flood forecasting models need to provide timely predictions but their spatial resolution can limit the effectiveness of the assimilation of satellite derived flood extents if limited changes of water depths do not imply significant changes in flood extension and if the model has not a sufficient resolution (Hostache et al., 2018).

In this work, a DA framework supported by heterogeneous observations coming from both local water level observations (i.e. stage gauges) and spatially distributed information gathered from satellite images - is proposed and tested. This research seeks to develop a more flexible DA scheme that may value all available sources of observations for distributed flood modelling updates. The aim of this work is to mitigate flood prediction uncertainties by combining heterogeneous data and an integrated topographic-hydrologic-hydraulic modelling approach, while maintaining inundation forecasting robustness, scalability and numerical stability. In achieving this goal, novel scientific advances and technical challenges of EO-driven DA approaches for flood prediction are investigated and in particular: A methodology for updating the state variable from multiple local Stage Gauges (SG) observations propagating the state variable corrections instead of applying localization in a hydraulic model for distributed flood routing in floodplain domains; the gathering of spatially distributed water level observations by means of flood extension processing and detection from satellite images, also adopting GIS-based algorithms for overcoming the issues of the different resolutions between the ensembles of the flood extents retrieved from the satellite derived images and the ones

generated from the hydraulic model simulations. This work conceptualizes and tests a framework for updating state variables of a Quasi-2D hydraulic model adopting the Ensemble Kalman Filter (EnKF) method to take advantage of observations gathered from heterogeneous sources. The Tiber river basin in central Italy is selected as case study that was recently the subject of flood events at the meso-scale level (approximately 100 km$^2$ of flood-prone domain), to investigate on improved flood modelling performances.

The paper is organized as follows: Section 2 describes the adopted hydrologic, hydraulic, and DA modelling methodologies. Section 3 illustrates the case study, the available data and the proposed DA implementation procedure. Section 4 discusses case study results, while section 5 provides conclusive remarks underlining advantages, limitations and suggested future developments of this research.

## 2    Methods

### 2.1    Hydrologic and 2D hydraulic modelling

The physically based Quasi-2D hydraulic model (FLO-2D, O'brien et al., 1993) was selected for flood wave routing and propagation. In fact, the regular grid mesh of its computational domain and the open format of the input and output files make the model simpler to be integrated in a Data Assimilation framework. The model solves the differential form of the dynamic wave approximation of the de Saint Venant equations with a central, finite difference numerical scheme. The numerical solution is applied along the river flowpath for in-channel 1D flood wave routing and for out-of-channel unconfined flood propagation considering 8 potential flow directions in a bi-dimensional (2D) domain. Channel and floodplain grid cells are assigned an absolute elevation, defining the floodplain and channel top surface topography. Channel conveyance capacity is considered by assigning to each cell a cross section with top banks associated to the corresponding floodplain cell elevation. The channel-floodplain flow exchange is simulated for taking into account over-bank and return flows within the riverine system. River and floodplain bridges, culverts, levees and any obstruction within the simulation domain are simulated in FLO-2D by means of rating curve, width and areal reduction factors. Two main boundary conditions were defined for the application of the Quasi-2D model: a) the floodplain domain extent; b) the hydrologic forcing from upstream for both the main river stem (i.e. source node) and for the tributaries.

The upstream and tributary flow boundary conditions b) are derived by stage gauge observations (where available) or from the application of a rainfall-runoff model (considering rainfall observations are available at river basin scale). A parsimonious hydrological model tailored for ungauged basins was selected to simulate the hydrologic forcing following Grimaldi et al. (2012) approach. This rainfall-runoff approach is based on the application of the geomorphic characterization of the Istantaneous Unit Hydrograph (IUH) adopting the WFIUH method (Mesa and Mifflin, 1986). In the WFIUH the shape of the river basin response to the rainfall forcing is associated to rainfall drop residency time distribution. The WF distribution may be expressed estimating the flow paths and associated to each flow path the travel time to reach the outlet (Rodriguez-Iturbe and

Rinaldo, 1997). The WFIUH distribution is, thus, estimated by applying channel ($v_c(x)$) and hillslope ($v_h(x)$) velocities to their corresponding flow paths $L_c(x)$ and $L_h(x)$:

$$WFIUH(t) = \frac{L_c(x)}{v_c(x)} + \frac{L_h(x)}{v_h(x)} \qquad (1)$$

165

The WFIUH distribution can be estimated using the DTM as main input information and applying terrain analysis algorithms for river basin hydrologic processing (pit removal- Jenson and Domingue, 1988-, flow direction, flow accumulation - Tarboton et al., 1991) to estimate flow paths at the basin scale. Channel velocities are considered constant according to Grimaldi et al. (2012). The hillslope velocity distributions $v_h$ are calculated according to NRCS (1997) as a function of the local slope and land use (Haan et al., 1994; McCuen, 2009). The adopted runoff modelling approach also considers distributed rainfall input and related infiltration losses using the SCS-CN method (Cronshey, 1986). Input rain gauge observations are interpolated using the Thiessen polygon methodology to properly assess distributed rainfall input for the hydrologic model (Thiessen, 1911).

## 2.2 Data Assimilation (DA) framework

A scheme of the whole DA framework with the reference of the related sections is illustrated in Figure 1. The Ensemble Kalman Filter method (EnKF- Evensen, 2003) was selected for DA application on the proposed 2D hydraulic modelling approach. EnKF, widely used in literature for DA, was selected for its efficiency in dealing with the significant non-linear flood dynamics (Reichle et al., 2002). The EnKF model is a sequential DA method that estimates the model state at time $t+1$ ($h_{t+1}$) based on the observations at the time steps in which they are available. The DA process is characterized by two steps: the forecast step and the updated step, whose variables will be represented respectively with the superscript $-$ for forecasting and $+$ for updating. The method is based on ensemble generations: the forecast (a priori) state error covariance matrix $P_{t+1}^-$ is approximated propagating the ensemble of the model states, according to the model errors expressed as a noise term $w_{t+1}$, from the previous time step; at the same time, an ensemble of observations $y_{t+1}$ at each update time is generated according to their error distribution introducing the noise term $\eta_{t+1}$. The updated probability density function ($pdf$) of the model states is given by a combination between data likelihood and forecast $pdf$ of the model states by means of Bayesian update. Specifically, the posterior estimate of the $i$-element of the ensemble $h_{t+1}^{i+}$ is calculated using the observation $y_{t+1}^i$ performing a linear correction with the Kalman filter to the forecast state ensemble members:

$$h_{t+1}^{i+} = h_{t+1}^{i-} + K_{t+1}[y_{t+1}^i - (H(h_{t+1}^{i-}, \theta) + v_{t+1}^i)] \qquad K_{t+1} = \frac{P_{t+1}^- H_{t+1}^T}{H_{t+1} P_{t+1}^- H_{t+1}^T + R_{t+1}^y} \qquad (2)$$

where $K_{t+1}$ is the Kalman gain matrix, $H(...)_{t+1}$ is a propagator relating the state variables to the measured variables and provides the expected value of the output given the model state, $v_{t+1}^i$ is the sample of the observation errors, $R_{t+1}^y$ is the variance of the observation error.

The performance of the ensemble forecast is influenced by the spread of the ensemble (Murphy, 1988; Anderson, 2001), but also by the ensemble size. The size has to be sufficiently large to represent a statistically significant sample, but at the same time

it has to be computational efficient considering the purpose of the application (e.g. near-real time updating of a flood model). In this work, the approach proposed by Anderson (2001) was selected. Therefore, the optimal ensemble size was selected to reach to a Normalized RMSE Ratio (NRR) equal to one.

The EnKF method application for the proposed Quasi-2D distributed hydraulic model was developed as follows. The state variable $h_{t+1}$ is associated to the water depth in a specific point of the computational floodplain domain. In case the observation is a stage gauge measurement, the state variable position is determined by identifying the closest channel cell. The correction is then applied also to the closest floodplain cells and propagated upstream and downstream as illustrated in Section 2.2.1.1 . In case the observations is gathered from a satellite image, the EnKF method is applied to both the channel and the floodplain cells for the entire computational domain as illustrated in Section 2.2.3.2 . The model error $w_{t+1}$ is estimated considering the uncertainties related to the input forcing $I_{t+1}$ and the model parameters as explained in Section 2.2.1.2. The observation $y_{t+1}$ is a water depth value gathered indirectly by the sensor. For this reason, the observation transition operation $H$ introduced in Eq. (2) is an identity matrix, being a direct relationship between state variables and observations.

### 2.2.1 Model updating

#### 2.2.1.1 Stage gauges observations

The application of the EnKF adopting one or more point measurements as observations for updating the state variables of a physical model has been deeply studied in scientific literature. Localization is a widespread method aimed to both reduce/avoid spurious (unphysical) forecast error correlations and reducing the dimension of the state vector (thus reducing the computational time) in computationally heavy models (Anderson, 2007; Hunt et al., 2007). Many localization methods include distance-based approaches for specifying the area of influence of an observation with a user-specified distance (Ott et al., 2004;Sakov and Bertino, 2011). Alternatively, there are adaptive localization methods (Anderson, 2007;Bishop and Hodyss, 2009 ) aimed to remove spurious correlations if distance-based localization is critical because of the model structure (Rasmussen et al., 2015), for example using a 'hierarchical ensemble filter', where a portion of ensemble filters is used to detect sampling error (Anderson, 2007). Localization can be applied assimilating independent local sub-domains (domain localization Ott et al., 2004), or multiplying the covariance of the forecast error by a Gaussian shaped correlation, that can be developed with a distance-based method (covariance localization, Houtekamer and Mitchell, 1998) allowing to assimilate observations "serially" (Tippett et al., 2003). Alternatively, Observation Localization (Hunt et al., 2007) applies the inverse of the above-mentioned distance based correlation to the covariance of the observation error (Observation Localization, Hunt et al., 2007) and demonstrated to have better performances than the covariance localization in some applications (Whitaker and Hamill, 2002). García-Pintado et al. (2015) is among the few cases in scientific literature in which localization (specifically Observation Localization) is applied to a 2D hydraulic model. The authors took in to account the physical connectivity of flows proposing a novel distance-based metric approach that consider channel network distance (instead of only Euclidean distance) for weighting the covariance of

the observation error and improved their model forecast skill as respect to the global filter and the traditional Observation Localization filter based on the Euclidean distance weight. Observation localization can be applied considering absolute water levels (as respect to the average sea level) or water depths (as respect to the terrain elevation) However, river water depths can dramatically change among contiguous cells of the hydraulic domain for example moving from a channel to a floodplain cell or because changes of the local geometry (e.g. cross section shape). In fact, usually stage gage measurements are located under hydraulic structures such as bridges, where the geometry of the cross section (that can be reshaped to be adapted to the bridge geometry) can have high differences as respect to the surrounding natural cross sections. Therefore, the localization techniques should be better aplied to absolute water levels. For this work, an observation localization technique (García-Pintado et al., 2015) applying a weight to the error covariance (i.e. a distance metric based also on a channel network distance) was implemented. A fifth order polynomial along-channel distance-based weighting function (Gaspari and Cohn, 1999) to correct the observation-error covariance matrix corresponding to a local analysis domain was applied. However, even changing the scale length of the correlation function, instability issues were encountered when updating the water levels far from the observation location especially in those areas with higher channel slope because the changes of terrain elevation from upstream to downstream. Therefore we proposed a simplified methodology aimed to assimilate observations at stage gague locations and propagate water depths correction (the difference between the posterior and the forecast state variables) for the surrounding channel and floodplain cells. The along channel upstream and downstream water level correction is performed applying a distance-based gain function adopting an approach similar to Madsen and Skotner (2005):

$$g(x_k) = A \cdot exp\left(-\frac{1}{2}\left(\frac{g'(x_k)}{1/3}\right)^2\right) \tag{3}$$

where $g(x_k)$ is the gain assigned to the $k$-cell, $A$ is the gain amplitude (assumed equal to 1), and the $g'(x_k)$ term is expressed as:

$$g'(x_k) = \begin{cases} \frac{x_{obs}-x_k}{x_{obs}-x_{uc}}, & x_{uc} \le x_k \le x_{obs} \\ \frac{x_k-x_{obs}}{x_{dc}-x_{obs}}, & x_{obs} \le x_k \le x_{dc} \end{cases} \tag{4}$$

where $x_{obs}$, $x_k$, $x_{uc}$, $x_{dc}$ are the linear coordinates along the channel of respectively the cell with the observation, the $k$-cell to be updated, the upstream and downstream bounds for the gain function. The last two terms allow to consider how far the updating could be inferred to correct the flood water profile. The gain function also allows to inject into the DA more than one observation for the same time step. The bounds of the gain for a $k$-cell are limited by the position of the closest stage gauge cells. Figure 2 provides a scheme of the adopted channel and floodplain model updating, depicting the propagation of the gain function upstream and downstream as respect to the observation point. The same correction at the k-cell is then assigned to the floodplain cells closest with a distance measure along the flow path. Furthermore, in order to properly assimilate more than one stage gauge observation, the channel segment (and its floodplain) that fall between two different simultaneous stage observations, are updated weighting the observation values by a multiplying factor expressed as the inverse of the distance

between the observation and target channel cells. The water level correction for the $k$-cell $\Delta H(x_k)$ is given by the following expression:

$$\Delta h(x_k) = \frac{\Delta h(x_{obs,u}) \cdot g(x_{k,u}) \cdot \frac{1}{x_k - x_{obs,u}} + \Delta h(x_{obs,d}) \cdot g(x_{k,d}) \cdot \frac{1}{x_{obs,d} - x_k}}{\frac{1}{x_{obs,d} - x_{obs,u}}} \tag{5}$$

where $\Delta h(x_{obs,u})$ and $\Delta h(x_{obs,d})$ are the water level updates respectively in the upstream and downstream stage gauges, $g(x_{k,u})$ and $g(x_{k,d})$ are the gains relative respectively to the upstream and downstream observation, $x_{obs,u}$ and $x_{obs,d}$ are the linear coordinates along the channel of respectively the upstream and downstream cells of the observation. When the gain function is propagated upstream and the water level correction is positive, a water profile counterslope may be inferred, causing a numerical instability issue in the hydraulic model. To avoid this issue, a further condition was imposed: the absolute water level in the channel cell $h_{abs}^+(x_k)$, cannot be lower than the adjacent downstream channel cell $h_{abs}^+(x_{k+1})$:

$$h_{abs}^+(x_k) = \begin{cases} h_{abs}^-(x_k) + \Delta h(x_k), & h_{abs}^+(x_k) \geq h_{abs}^+(x_{k+1}) \\ h_{abs}^+(x_{k+1}), & h_{abs}^+(x_k) < h_{abs}^+(x_{k+1}) \end{cases} \tag{6}$$

The proposed simplified updating approach allows to remarkably reduce the dimension of the state vector, considering only the locations with observations, therefore it is expected to avoid filter convergence issues while pursuing acceptable computational efficiency. The model updating procedure is invoked at each time step when one or more observations become available. The hydraulic simulation is stopped saving distributed floodplain water levels and volume conservation to binary files. Then, the EnKF is applied and the water depth corrections are applied to update model states in the binary files.

#### 2.2.1.2 Satellite image observations

The assimilation of flow depths derived from satellite image processing is developed following 3 main steps:

1. Flood detection from satellite image(s);

2. Comparison of the flood extent detected from the satellite image with the ensemble of flood extents simulated by the hydraulic model;

3. Derivation of the water depth distribution related to the satellite image starting from the ensemble of the water elevation distributions of the hydraulic model.

1) The proposed methodology aims to be applicable to both multispectral and SAR images to increase the take advantage of all available satellite observations of a flood event. Considering multispectral images suffer of significant limitations due

to cloud cover and light conditions, the Modified Normalized Difference Water Index ($MNDWI$) proposed by Xu (2006) is applied. The $MNDWI$ is expressed as:

$$MNDWI = \frac{\rho_{bg} - \rho_{bm}}{\rho_{bg} + \rho_{bm}} \tag{7}$$

where $\rho_{bg}$ and $\rho_{bm}$ are the reflectance indices of respectively the Green and Mid Infra Red (MIR) bands. For SAR images, the image histogram thresholding methodology is implemented following Brivio et al. 2002.

2) The satellite detected water extension is, then, compared with the flood extension ensemble simulated by the hydraulic model ($HM$) at the time step of the satellite image's acquisition date. In order to avoid the impact of resolution issue on the
comparison, the simulated flood raster is downscaled at the same resolution of the satellite image by following this procedure:

–  The water surface elevation is interpolated at the satellite image resolution by applying the Kriging method (Matheron, 1969; Oliver and Webster, 1990) using the maximum floodplain extent polygon ;

–  The interpolated Water Surface Elevation (WSE) is intersected with a high resolution DTM to flag positive values as potentially flooded.

The two raster flood maps (Water extension from $SI$ and from $HM$) are, then, quantitatively compared by applying the measure-of-fit F-index (Horritt and Bates, 2001):

$$F = \frac{A}{A + B + C} \tag{8}$$

For the generic $k$-cell pertaining to the hydraulic modelling domain, the satellite-derived indirectly observed water depth $h_{o,t}^k$ at the time $t$ , is expressed as:

$$h_{o,t}^k = h_{m,i_1,t}^k \cdot \frac{F_{i_1}}{F_{i_1} + F_{i_2}} + h_{m,i_2,t}^k \cdot \frac{F_{i_2}}{F_{i_1} + F_{i_2}} \tag{9}$$

where $F_{i_1}$ and $F_{i_2}$ are the two best fitting flood maps from the ensemble of the $HM$ compared to the flood extent from the $SI$, $h_{m,i_1,t}^k$ and $h_{m,i_2,t}^k$ are their related the flow depths of the $k$-cell at time $t$.

### 2.2.2   Model errors

The uncertainty related to model errors are numerically managed within the proposed DA by perturbing:

–  the hydrologic forcing input given by the upstream static sensors and the rainfall-runoff modelling output;

–  the hydraulic model parametrization associated to channel roughness expressed by the distributed Manning coefficients.

In both cases the flow discharge values at time $t$ of the $s$-input for the $i$-element of the ensemble are expressed using a similar approach to García-Pintado et al. (2013). The matrix of inflow errors is expressed as the composition of a temporally correlated error and an heteroscedastic error, whose variance is proportional to the flow value at time $t$. The temporally correlated error for the $i$-ensemble member, $q^i_{s,t}$, evolves individually according to the expression proposed by Evensen (2003):

$$q^i_{s,t} = \rho_t \cdot q^i_{s,t-1} + \sqrt{1 - \rho_t^2} N(0, R_s) \tag{10}$$

where $\rho_{s,t}$ is a temporal autocorrelation coefficient and $N(0, R_s)$ is a white noise with a given variance $R_s$. The temporal autocorrelation coefficient between two time steps $t$ and $t+\Delta t$ is imposed as a function of $\Delta t$ and a specified time decorrelation length $\tau$ (Evensen, 2003) as follows:

$$\rho_t = e^{-\frac{\Delta t}{\tau}} \tag{11}$$

the variance $R_s$ of the white noise is imposed equal to 1 (Evensen, 2003). This spatially indipendent value is reasonable for $SG$-derived flows which should not include spatial correlation errors, since errors and uncertainties in $SG$ measurements should not depend on the gauge position. On the other hand, the variance $R_I$ related to an input derived form the hydrological model should be depend on the distance between the locations of the other inflows, considering that the precipitation field is the main input forcing of the hydrologic model and one of the most impacting factor in flood mapping uncertainties for hydrologic-hydraulic modelling (Annis et al., 2020). Therefore, the spatial correlation $R_I^{x,y}$ between two inflow errors $x$ and $y$ ie expressed with a Gaussian-decay correlation model (García-Pintado et al., 2009):

$$R_I^{x,y} = e^{-\frac{1}{2}\frac{d_{x,y}}{\theta}} \tag{12}$$

where $d_{x,y}$ is the distance between the $x$ and $y$ locations and $\theta$ is a spatial correlation coefficient.

As proposed by García-Pintado et al. (2013), the matrix of the heteroscedastic error is obtained as the element-wise product of the above mentioned temporally correlated error with the following factor:

$$\sigma^i_{s,t} = \sqrt{(\alpha_s \cdot Q^{os}_{s,t})^b} \tag{13}$$

where $Q^{os}_{s,t}$ is the $SG$-derived or simulated streamflow value by the $s$-input at time $t$, $\alpha_s$ is the coefficient of variation related to the uncertainty of the discharge, $b$ is an heteroscedasticity factor. Equation 13 infers the intuitive principle that high discharge values are more uncertain than low values. The resulting heteroscedastic error is then applied to the term $\gamma \, Q^{os}_{s,t}$ where $\gamma$ is a multiplicative bias factor.

The uncertainty related to discharge observations gathered from static sensors (SG) is the sum of two components (Clark et al., 2008): the estimation of the water level from the static sensor reading ($EWL$); the conversion of the water level into discharge

using the fluvial cross section rating curve ($ERC$). In this work, the coefficient of variation related to the static sensor was set to $\alpha_{SG} = 0.1$, where $\alpha_{EWL} = 0.1$ (Weerts and El Serafy, 2006; Clark et al., 2008; Rakovec et al., 2012b) and $\alpha_{ERC}$ is considered negligible as respect to the $\alpha_{EWL}$ (Baldassarre and Montanari, 2009). The coefficient of variation related to the input provided by the hydrologic model ($\alpha_I$) can be derived from a validation analysis of the hydrologic model calculating the distribution of the simulated flow errors. For both uncertainties related to $SG$ and $I$, $\gamma$ and $b$ values were imposed equal to 1.

In addition to the uncertainty due to the hydrologic forcing, the uncertainty related to the channel roughness is also considered as follows (Clark et al., 2008; McMillan et al., 2013):

$$p_s^i = p_s + U(-\epsilon_p \cdot p_s, +\epsilon_p \cdot p_s) \tag{14}$$

where $p_s^i$ is the perturbed model parameter for the $i$-element of the ensemble, $p_s$ is the calibrated model parameter and $\epsilon_p$ is the fractional parameter error.

To avoid potential systematic underestimation of the model covariance error due to the limited ensemble size (overconfidence in prior estimates), the inflation method (Anderson, 2001) was implemented. The percentage of increment of the ensemble forecast anomalies can be considered as constant (Anderson and Anderson, 1999; Whitaker and Hamill, 2002 ) or time variant (Ott et al., 2004), such as a ratio between a user-defined standard deviation as respect to the forecast standard deviation error 350 (Rasmussen et al., 2015), or between the forecast and the updated forecast standard deviation (García-Pintado et al., 2015). In this work we adopted the first approach where, according to Evensen (2003), each i-element of the forecast state variable $h_t^{i-}$ is expressed as follows:

$$h_t^{i-} = \lambda(h_t^{i-} - \overline{h_t^-}) + \overline{h_t^-} \tag{15}$$

where $\lambda$ is an input inflation parameter imposed equal to 1.01 (Evensen, 2003), $\overline{h_t^-}$ is the average value of the state variable 355 ensemble at time $t$

### 2.2.3 Observation errors

#### 2.2.3.1 Errors related to stage gauge observations

The errors associated to observations of the $SG$ within the floodplain domain are considered by performing a perturbation 360 of the observed value using a similar approach adopted for perturbing the input flow from stage gauges, as combination of a temporally correlation error (Eq. 10) and a heteroscedastic error expressed as

$$\sigma_{SG,t}^i = \sqrt{(\alpha_{SG} \cdot h_{SG,t}^{obs})^b} \tag{16}$$

where $h_{SG,t}^{obs}$ is the observed water level value by the static sensor at time $t$. In this case, there is no error due to the rating curve application, considering the water level observations are directly compared to the simulated ones, therefore the coefficient of variation $\alpha_{SG}$ is assumed equal to 0.02 m (Schmidt, 2002; Pappenberger et al., 2006).

### 2.2.3.2 Errors related to satellite image observations

The procedure adopted for deriving water depth distributions from satellite images is affected by a series of errors that must be taken into account and in particular:

— *Error in the water surface detection from satellite images* this error is due to the water detection technique that could overestimate or underestimate the flood extension. Both multispectral and SAR image processing for water extent mapping require a threshold to apply in respectively the Water Index and the backscatter coefficient. Literature values of these thresholding values could lead to inaccuracies considering optimal threshold values are usually case study or event specific; therefore, a perturbation of the threshold value is performed by adopting a normal distribution with zero mean and a standard deviation derived from literature values (Pierdicca et al., 2009).

— *Error of the water surface extraction from the simulated WSE of the hydraulic model.* This error is mainly due to the vertical error of the DTM that is used in the water surface elevation interpolation procedure. The generic $i$-DTM of the ensemble is perturbed generating a vertical error with a normal distribution characterized by a zero mean and a variance that is uniformly distributed between 0 and 0.3 m - $U(0,0.3)$ - according to literature values (Hodgson and Bresnahan, 2004; Leon et al., 2014; Brouwer et al., 2017). Considering the proposed normally distributed independent errors does not take into account the spatial continuity of the elevation data (Raaflaub and Collins, 2006; Heuvelink et al., 2007), a GIS algorithm for inferring spatially autocorrelated errors is applied. A Correlation Distance Error ($CDE$) equal to 100 m is applied according to Li et al. (2011), Livne and Svoray (2011), Mudron et al. (2013), Leon et al. (2014). The above-mentioned GIS algorithm includes the following steps: 1. Generation of a raster (NR) of random values with a normal Gaussian distribution ($\mu$=0, s=1) for the entire extension of the DTM; 2. generation of a raster (SR) with the average of the NR values within a neighborhood equal to CDE; 3. Creation of a error distribution raster (Err) dividing the SR raster by its spatially averaged standard deviation and multiplying the result for the adopted variance U(0,0.3); 4. Addition of the Err raster to the original DTM.

— *Error of the water depths derived from the ensemble of hydraulic modelling.* Eq 9 assumes a linear relationship between water depth values of two hydraulic profiles and the weight associated to their relative F-indexes expressing the comparison with the observed water extension from $SI$. The application of this weighted mean of the simulated water depths could lead to an inaccuracy on the vertical estimation of the water depths, especially at the boundaries of the two different simulated flood extents. The perturbation error due to the profile derivation for the $i$-element of the ensemble

and the generic $k$-cell is expressed as a random uniform noise:

$$err_{PD}^{i,k} = U(c \cdot \Delta h_{12}^k, +c \cdot \Delta h_{12}^k) \tag{17}$$

Where $\Delta h_{12}^k$, is the water level difference at the $k$-cell of the two best fitting hydraulic simulations (see Eq. 9); $c$ is a coefficient ranging between 0 and 1, considering that the gentle terrain slopes in floodplains limits the error of water depths derivation in an interval smaller than $\Delta h_{12}^k$.

## 3   Case study: available data and DA implementation

### 3.1   The Tiber River in central Italy

The selected case study is represented by the Tiber river upstream of the city of Rome (Figure 3). The fluvial transect goes from the village of Orte Scalo to the northern boundary of the city of Rome corresponding to the Castel Giubileo dam. The entire floodplain domain of the Tiber Orte- Castel Giubileo transect has an extension of 5881 km$^2$, with a main tributary represented by the Nera River (drainage area of 4180 km$^2$) and 15 minor ungauged tributaries draining into the selected fluvial domain. The Tiber river at the upstream Orte boundary section has a drainage area of 8400 km$^2$, while at the downstream end of Castel Giubileo the drainage area is 14850 km$^2$ (total Tiber river basin catchment area at the Tyrrhenian sea outlet is approximately 17400 km$^2$). The floodplain domain is mostly characterized by agricultural use, but major road and railway infrastructures were developed connecting several urbanized areas along the Tiber floodplain with four main towns Orte Scalo, Fiano Romano, Monterotondo and the northern part of the city of Rome that have been the subject of frequent floods in January 2014, November 2012, November 2010, and November 2005, causing damages to buildings, roads and bridges. This floodplain domain has also a strategic importance for the flood risk mitigation of the city of Rome considering flood volume accumulation in this domain determine a significant flood peak attenuation that propagates through the historical city center. Understanding, monitoring and predicting flood scenarios in this fluvial domain is crucial for protection the socio-economic and cultural assets of the Italian capital city. The city of Rome EWS strictly relies on the flood modeling predictions of the selected area.

### 3.2   Parametrization of the flood forecasting model

#### 3.2.1   Topography and hydrologic modelling

Topographic data to represent the morphology of the selected Tiber river subbasin domain were gathered from the Tinitaly 10 meter resolution DTM (Tarquini et al., 2012) for supporting the hydrologic modelling. Rainfall time series for rainfall-runoff modeling were gathered from 94 rain gauges with a temporal frequency ranging from 1 to 15 minutes. SCS infiltration method parametrization used 4th level Corine Land Cover dataset gathered from the *Istituto Superiore per la Protezione e la Ricerca Ambientale* (ISPRA) repository, with ancillary data for the lithology and permeability layers gathered from *Autorità di Bacino Distrettuale dell'Appennino Centrale*. The river basin terrain analysis procedure needed to provide WFIUH hydrologic modelling input parameters used a value of 1 $km^2$ to define stream network source cells, a constant parameter that adequately

matched the fluvial network extension observed from aerial images of the basin. WFIUH kinematic parameters were calibrated using 4 small gauged basins (Naja, Niccone, Puglia, Sovara) estimating channel flow velocities equal to 2 m/s and distributed hillslope flow velocities in the range 0.01 to 0.1 m/s.

### 3.2.2 2D hydraulic modelling

The bathymetry needed to represent the channel conveyance capacity in the hydraulic model was derived by interpolating available surveyed cross sections. The continuity of the channel-floodplain topographic domain was obtained by using the available high resolution LiDAR DTM (1 meter resolution ), gathered from *Ministero dell'Ambiente e della Tutela del Territorio e del Mare*, that covers most of the selected Tiber river floodplain area. Where LiDAR was not available, the 5 meter resolution DTM from *Regione Lazio* was used. The floodplain grid cell resolution used for 2D flood unconfined flow routing was set equal to 150 meters to produce computationally efficient model runs. The consistency of the coarse resolution hydraulic model was validated by matching flood simulations with available observations from three recent inundation events (November 2005, November 2010, November 2012). Available flood observations were also used to calibrate the roughness parametrization. A constant Manning coefficient for the channel equal to $0.04m^{-1/3}$ and variable Manning parameters for the floodplain, classified using Corine land cover classes, were calibrated. The 2D hydraulic model validation was performed by comparing simulations with available flood stage time series from seven gauges that were used as floodplain control sections. This also allowed to verify the consistency of the numerical representation of major urban features of the study domain. Flow/stage rating tables available for the main bridges and weirs gathered from the *Centro Funzionale regionale del Lazio* were used to finalize the 2D hydraulic model validation.

### 3.3 DA implementation

The ensemble size for the application of the EnKF model was defined, according to Anderson (2001), by analyzing the available stage gauges and the selected flood events. The optimal ensemble size was set equal to 40.

To define the temporal autocorrelation error (Eq. 11), a time decorrelation length $\tau$ equal to 3 days was imposed, while a spatial correlation coefficient $\theta$ was assumed equal to 60 km. García-Pintado et al. (2013) considered these values as representative of a spatially distributed or semidistributed model, that ingests continuous rainfall field inputs after being calibrated with previous flood events.

The hydrologic model is affected by different sources of uncertainties: the structural uncertainties, given by the simplification of the modelled physical processes (e.g. we adopted a WFIUH approach, neglecting groundwater flow, mud and debris flow), the input uncertainties (given by the rainfall values and antecedent soil moisture conditions), and parametric uncertainties due to the inaccuracy of the model calibration). These sources of uncertainty should be considered separately. For example input rainfall uncertainty from rain gauges can be estimated considering quantitative precipitation ensembles (Clark and Slater, 2006), such as Sequential Gaussian Simulations (Goovaerts et al., 1997; Rakovec et al., 2012a). Precipitation ensemble generated with NWPs can than be coupled with hydrologic models to improve flood forecasting (Jasper et al., 2002;Sorooshian et al., 2008). In this work we adopted a simplified procedure taking in to account all the modeling uncertainties considering the

frequency distribution of the errors between the observed and simulated flow values obtained by the calibration and validation of four small tributaries of the Tiber river basin in past flood events. From the validation of the hydrologic model, the frequency distribution of the relative flow errors was characterized by a mean equal to zero and a standard deviation equal to $0.28 \cdot Q^{os}$, thus $\alpha_I$=0.28 (Eq. 13).

The application of Eq. 14 to consider channel roughness parametrization uncertainties resulted in a $p_s = 0.04 m^{-1/3}/s$ (according to the hydraulic model calibration) with $\epsilon_p$ assumed equal to 0.125. This limits the Manning channel value variations between a minimum of 0.035 and a maximum of 0.45 $m^{-1/3}/s$. The floodplain roughness uncertainty was considered less significant considering that the governing factor for the Tiber floods in the selected domain is the volume, while minor specific urban features and singularities characterized the selected flood events. It is also noted that, for the selected events, the flow is conveyed by the channel for most of the simulation time.

A Landsat 7 image (acquisition date: 14/11/2012 - 09.43) was processed using Eq. 7 to extract the observed flood extension. Landsat 7 products are affected by minor corruptions due to a failure of the satellite Scan Line Corrector (Scaramuzza and Barsi, 2005) that create some data void (i.e. empty stripes) in the water mask. These irregularities were analyzed and interpolated, allowing to overcomes this issue, and define a correct delineation of the flood extension, clearly visible from the available Landsat 7 image. Figure 4 shows the resulting flood detection map. The Landsat 7 image extension covers only a portion of the selected study domain. The satellite image acquisition time is consistent with peak flow conditions within these two gauging stations (Figure 4). A threshold value equal to 0 for MNDWI was chosen (Xu, 2006). To apply the EnKF method, the variance of the random noise related to the threshold value of MNDWI was set equal to 0.2.

## 4    Results and discussion

Three sets of DA scenarios were selected: 1. stage gauge observations, 2. satellite image observations 3. both SG and SI. In case of assimilation from $SG$-derived observations, two sub-scenarios were implemented: the simultaneous assimilation of 4 stage gauges (ASS 4SG) and the assimilation of one upstream stage gauge, Ponte Felice, (ASS 1SG). This latter were introduced to analyse how the model performance vary downstream far from the observation location.

The 2005, 2010 and 2012 floods were selected for applying the DA framework using stage gauges observations. In case of scenarios 2 and 3, the Landsat 7 image, described in Section 3 was assimilated for the 2012 flood event.

### 4.1    Assimilation of stage gauge observations

Figure 5 shows the comparison of observed and simulated water level time series at Stimigliano, Nazzano and Ponte del Grillo stations (respectively 16.6, 48.2 km and 59.1 km far from the upstream Ponte Felice station) for the 2005, 2010 and 2012 flood events. The first two flood events are characterized by multiple peaks whose rising and recession curves are not properly represented by the open-loop (OL) simulation. This limitation is probably due to the coarse resolution of the flood model. In

fact, wetting and drying phenomena along preferential flow pathways are usually influenced by the micro-topography of the domain, that are better represented in higher resolution models (Nicholas and Mitchell, 2003; Neal et al., 2011). The simultaneous assimilation of four stage gauges (ASS 4SG) is able to overcome this issue and the spread of assimilated ensemble at each stage gauges is much lower than the one of the OL simulation because of the small error associated to the stage gauge observation as illustrated in Section 2.2.3.1. On the other hand, the assimilation of the only upstream stage gauge (ASS 1SG) provides a slight benefit, as respect to the $OL$ simulation, at the closest stage gauge (Stimigliano), mostly for high values of flow depths (e.g. at the peak of the 2012 flood event), while it does not imply any substantial changes downstream, where local inflow conditions and terrain geometry completely attenuate the upstream water levels corrections. Table 1 shows the performance indexes for OL, ASS 4SG and ASS 1SG simulations considering the stage gauge observations as true values, given their low uncertainty.Bias is expressed as ratio of the sum of the observed and simulated water levels. The scenario ASS 4SG improves the prediction of the water levels in terms of NSE, R and Bias performance indexes (Table 1), while the performances related to the scenario ASS 1SG decrease gradually downstream as respect to the assimilated gauge observation. Note that the ASS 1SG performances are even slightly worse than $OL$ in Nazzano and Ponte del Grillo. This is due to the fact that propagation of corrections related to far observation locations can be counterproductive (Giustarini et al., 2011). Bias for OL and both ASS simulations tend to increase above 1 in the downstream stage gauges because of their overestimation of the water levels, especially in the recession limb (that is less important for EWS). For the same reason, for all the simulations, the more the flow is far from to the upstream inflow, the more the $R$ coefficient tends to decay, but in case of $ASS4SG$ simulation, this decay is mitigated.

Figure 6 shows the channel water depths profiles for three different time steps (correspondent to three different position of the peak flow along the channel) for $ASS4SG$ (left panles), $ASS1SG$ (rigth panels) and ($OL$ both right and left panels) simulations for the November 2012 flood event. The $ASS4SG$ scenario shows improvements in terms of reduction of ensemble spread in most of the channel domain as respect to the $OL$ simulation, even if the the adopted gain function illustrated in Eq. 3 attenuates the correction mostly in the downstream part of the domain, far from the flow observations. Note that the assimilated water depths right upstream the Stimiglaino stage gauge (second black dot from left) because of the propagation of the water depth correction of the upstream stage gauge (Ponte Felice). In the $ASS1SG$ scenario, the corrections of the water depths a respect to the $OL$ simulation are almost negligible in the downstream part of the domain.

Spatial covariances of the assimilated flow depths (represented in Figure 7 for a portion of the computational domain) increase with simulation time during the rising limb for both assimilation scenarios (ASS 4SG and ASS 1SG), but ASS 4SG show lower values of covariance as respect to the ASS 1SG simulation especially close to the downstream SG location. This confirm that Quasi 2D hydraulic model is able to improve its performances mostly if multiple stage gauges observations are assimilated, since propagation of corrections are attenuated by downstream inflows and local geometry.

Figure 8 shows the impact of water level updating model ASS 4SG on the simulated flood extent. In Figure 8 (b) a simulated flood extension subset is shown considering the mean levels of the OL and ASS 4SG simulations for the November 2012 event.

The ASS SG flood extent simulations are on average 6.5 $km^2$ larger than the OL simulations. Major differences are located in the flat areas where flood extents are very sensitive to flow depth variations. In this Figure, the resolution of the flood extension of the hydraulic model was increased interpolating the water surface elevation at the the 5 meter resolution DTM by applying the Kriging method and filtering out the cells with lower resolution a respect to the 5 meter resolution DTM. This method can be,thus, accepted in case of a long persistence (e.g. several hours) of water levels at the same location, as in the selected case study. Inaccurate flood extent mapping are expected for small basins with low flood persistence. The application of this methodology for smaller basins should require an higher flood model and DTM resolution.

## 4.2  Assimilation of the satellite-derived flood extent observations

Figure 9 shows the observed and simulated flood hydrographs at Stimigliano, Nazzano and Ponte del Grillo stations for both the OL and ASS models for the 2012 flood event. The updated mean water levels at the SI acquisition time are slightly higher, and persist to stay higher than the OL simulation for few hours. The spread of the ensemble of the ASS simulation is significantly reduced in correspondence of the SI observation. This reduction is gradually damped until it completely disappears in approximately 8 hours. The improvement of simulation NSE and Bias for the SI assimilation case are not significant (Table 2), considering that the updating persists for only few hours, as shown in Figure 9.

Figure 10 a) shows the reduction of the water levels uncertainty and the correction of the mean value along the channel profile at the SI acquisition time. Note that the proposed methodology for gathering the indirectly observed Satelite-derived water depths allowed to have observations at each cell, therefore the EnKF is applied serially to the whole domain with positive depths values instead of only at the SG locations, thus avoiding increase of the water depth ensemble spread in cells far from the SG locations as showed in 6 . Reduction of the spatial covariances right after the SI acquisition time as respect to the OL simulation are showed in 10 b) and c). Covariances for OL are much lower than ASS SI mostly along the channel cells. Covariances of ASS SI simulation are also lower than the ones generated by the ASS 4SG and ASS 1SG observations at the same step (see the right panels of Figure 7) especially far from the SG locations. The adopted updating procedure allows to increase the flood extent of 4 $km^2$ at the time of the SI acquisition (see an inset of the flood extension in a flat area of the floodplain domain in Figure 11) leading to reduce the False Negatve rate by 7% while an increase of the false positve rate of only 1%. Note that the satellite derived flood extent is considered as true flood map. Despite the smaller water level observation errors given by the stage gauges observations as respect to the SI, the overall change of the mean flood extent between the OL and ASS SI simulations ( 14% ) does not show significant differences as respect the Stage gauge DA (9%).

Figure 12 shows the variability of Bias, Root Mean Square Errors (RMSE) and Standard deviation of the ensembles calculated starting from the from the time of the acquisition of the satellite image the comparing OL and ASS modelling results at Stimigliano, Nazzano and Ponte del Grillo stations. Improvements in terms of Bias and RMSE are significant for 20 hours after the SI acquisition, while the uncertainty reduction (i.e. the difference in the ensemble amplitude between the OL and ASS

SI simulations) persists for 8 hours. This behaviour suggests that observations right after the SI assimilation step (e.g. the SG observations) could benefit from the reduction of the model uncertainty for the whole computational domain.

## 4.3 Assimilation of both stage gauge and satellite observations

At the positions where the channel flow gauges are located, it is reasonable to expect that stage gauges observations support better model performances as compared to model updated based on satellite observations (see Figures 5 and 9). However satellite image observations provide spatially distributed information that are important for flood wave routing in complex domain, especially when flooding impact large unconfined domains. Obtaining multiple stage gauge observations and satellite images constitute the optimal support for DA application for EWSs.

To investigate the potential benefit of taking advantages of multiple heterogeneous distributed data sources of flood observations, we simulated the simultaneous assimilation of the four SG observations and the SI derived flood extent (ASS TOT). As expected, the water level time series at the stage gauges locations are similar to the ones of the disjoint ASS 4SG simulation (bottom panels of Figure 5) in terms of mean and spread of the ensemble (therefore plots of hydrographs at SG are not shown for the joint assimilation simulation). Specifically, improvements in performances as respect to the OL simulation are observed 570 in terms of increase of NSE (5 - 40%), increase of Pearson correlation (up to 12%) and Bias reduction up to 80%. (see Table 3)

Conversely, differences as respect to the 4SG assimilation are found far from the SG locations in terms of ensemble spread of channel water levels and spatial covariances (Figure 13) at the SI the acquisition time - a) and c) panels- and 2 hours later - (b) and d) panels. The spread of the ensemble of the water levels along the channel profile is reduced as respect to both the ASS 575 4SG assimilation and for the ASS SI assimilation. Moreover, covariance values of ASS TOT are smaller as respect to the ones of ASS SI and ASS SG simulations for the same time step (see Figure 7 panel c, Figure 10 panel c and 13 panel c). In fact, the combination of the model $pdf$ right before the SI acquisition (that is already narrowed by the previous SG assimilation) with the $pdf$ of the SI derived observed water depths allows to generate a water depth ensemble with a lower spread as respect to both the disjoint SI and SG assimilations. The joint assimilation of SG and SI observations (ASS TOT) has also a positive im-580 pact on the flood extent uncertainty (see Figure 14) if compared with the OL and disjoint SG and SI assimilation. Specifically, the ASS TOT simulation led to a reduction of the maximum flood extent variability for several hours after the SI acquisition as respect to the ASS SG and ASS SI simulations. Figure 14 shows also a consistent difference of the flood extent ensemble spread between the ASS 1SG and the ASS 4SG simulations, confirming that the propagation of the assimilation of a single stage gauge has a spatially limited effect (only for the cells around the SG location), while the joint SG and SI assimilation 585 improves the performances after the SI acquisition time for almost 15 hours.

## 4.4 Pros and limitations

The proposed modelling chain approach is affected by some limitations but also advantages that are summarised in this Section. Firstly, the adopted hydraulic model has a coarse spatial resolution (150 m cell size) and its performance can be considered acceptable for high magnitude flood events; however some limitations could raise in representing the flow patterns for low

magnitude events, where micro topography can have an important role in the flow propagation (Bates 2012). Further tests considering smaller domain with a higher resolution 2D hydraulic model are needed to verify the stability of the model updating, when water levels corrections are applied to cells characterized with smaller dimensions. Moreover, the model uncertainty can be considered quite large during the peak flows of the selected flood events with an amplitude of the ensemble of the water levels even equal to 2 meters (see Figures 9 and 10). More accurate models could not benefit from the assimilation of the

satellite image, if the indirect water level distribution derivation is affected by uncertainties larger than the one ones related to the forecasting model. Tables 1, 2 and 3 shows also some bias in the hydraulic model, mostly prominent in the recession limb of the hydrographs. This is a limitation, since the optimality of the DA techniques is realized if the observations and the models are not biased (Dee, 2005; Liu et al., 2012).Bias reductions and further improvement of the simulation can be done by updating the model inputs (i.e. the inflow hydrographs) and parameters (e.g. the roughness) using an augmented state vector

approach in the EnKF framework (Montzka et al., 2012).

    The simplified rainfall-runoff modelling allowed to generate input flow hydrographs very quickly according to the needs of a near-real time flood modelling purpose. However, the model can be considered appropriate only for small basins characterized by an impulsive response, for which the groundwater component can be neglected, complex topography and flow control structures are absent to avoid equifinality issues during the calibration/validation analysis (Beven, 2006). Furthermore the ap-

plication of the SCS-CN model at sub-daily time scale (Grimaldi et al., 2013) is a strong limitation and more advanced models should be preferred for reduce the model uncertainties.

    Despite the several measures adopted to prevent instability issues, instability can occur during the updating of the water levels from the stage gauges. For this reason, the model is tailored to remove from the ensemble the critical elements causing instability and generate new elements in order to keep the sample size constant. This measure can slow down the model, that should

be as fast as possible for a proper near-real time application. The instability issues that sometimes can occur could be due to the fact that FLO-2D model doesn't allow to update the flow velocities but only the flow depths. This limitation doesn't also enable to have control of the volume conservation, that is an important factor to verify the accuracy of the model simulation. The need of updating the model states each time the stage gauge observations are available can affect the efficiency of the model in terms of computational performance. The proposed model requires averagely 3.7 minutes for each simulation hour with a standard

laptop (processor: 4 core with 180 GHz each and 8 GB RAM). Alternative approaches such as the Asynchronous Ensemble Kalman filter (AEnKF, Sakov et al., 2010), allowing to ingest past observations over a time window, to update model state at a specific time step, could help reducing the times when the model is updated, even if each model updating require little higher costs to the computational time and storage requirements.

    A methodology for indirectly deriving the distribution of the water depths from the water footprint gathered from a satellite

image was proposed; this methodology is affected by a series of errors that were taken in to account for assigning a proper reliability to the observation related to the satellite image; this reliability is numerically lower than the one related to stage gauges observations, but at the same time, provides distributed information instead of the local ones given by the static sensors. The derivation of the water depths from the flood extent gathered by the satellite image was performed with a linear combination of the values given by the ensemble of the results provided by the hydraulic forecasting model. Since the latter has to be

updated by an observation that is indirectly derived by the model itself, this approach can be considered disputable; however, practically it demonstrated to not cause instability issues during the model updating, since the distribution of the flow depths is coherent with the model state variable and it has demonstrated to slightly improve the model performance. This approach can be considered as an hybrid methodology of two literature approaches that consider prognostic and diagnostic variables for assimilating satellite derived information (Hostache et al., 2018), taking advantage of the rapid flood detection algorithms adopted for direct assimilating the flood extent and, at the same time, directly retrieving the water levels that are prognostic variables, thus more straightforward to assimilate than the flood extent (Hostache et al., 2018).

During a flood event, the adoption, as observation, of a multispectral image, potentially corruptible by the cloud covering and sunlight is much less likely than the one with a SAR image; however, the proposed approach for the model updating can be applicable regardless the type of image as input observation. The satellite revisit time of the current satellite missions is still a strong limitation, since can be much higher than the travel time of small basins. Furthermore, usually multispectral SAR images require time for being processed. However, new satellite missions and also the combination of more constellations will considerably reduce the revisit time in the near future, allowing to have different images for the same area with a temporal frequency higher than the persistence of the model correction (8 hours). Moreover, recent automatic satellite image techniques for extracting the flood extension have been implemented in real time services for flood mapping (Martinis et al. 2015).

## 5    Conclusive remarks and future work

The proposed DA framework investigated on the opportunities and challenges of assimilating multiple sources of observations for improving the performances of near real time flood predictions in case of some real flood events, selected as case study. Specifically, stage gauge water level readings and satellite-derived flood extents were used for testing the proposed DA framework. We infer the following main conclusions:

- The assimilation of multiple (four) stage gauges significantly improved the flood model performances in terms of NSE, R Bias, also reducing simulated inundation extent uncertainties, however the spatial influence of assimilation is limited if only one SG observation is adopted ;

- The assimilation of distributed flood depths, indirectly retrieved from satellite-derived flood extent, provided slightly better modelling performances only few hours after the SI acquisition and allowed to reduce the water level and flood extent uncertainties for the whole computational domain.

- the simultaneous assimilation of SG and SI observation brought to a reduction of the water level and flood extent uncertainty for several hours as respect to the disjoint SI and SG assimilation after the SI acquisition time.

Future tests are needed, taking advantage of increasing availability of satellite-derived flood extents at higher temporal and spatial resolution, to discover the effective capacity of the proposed flexible multi-source DA framework to value a larger EO

data availability.

Moreover, the flexibility of the proposed model, to assimilate local and distributed observations, suggests its suitability to use other data sources, also gathered from informal observation systems. In particular, future work will allow to investigate on the use of crowdsourced observations to apply the proposed flood prediction framework in ungauged basins. Crowdsourc-
ing already proved to be an effective means to improve hydrologic (Mazzoleni et al. 2017) and hydraulic (Annis and Nardi 2019) modelling performances , but further work is needed to test the use of crowdsourced data in real time flood modelling approaches. Significant improvements are expected in the near future for improved weather predictions by valuing all data sources, especially affordable citizen-driven observations for developing countries that are the most vulnerable to hydro-extremes (Alley et al. 2019).

*Acknowledgements.* FN and AA acknowledge the support received by the WARREDOC center of University for Foreigners of Perugia through the WARREDOC-Fondazione ENI Enrico Mattei (FEEM) research agreement. FN acknowledges the support received by the Southeast Environmental Research Center in the Institute of Environment at Florida International University.

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

# TABLES

| Event | Station | NSE | | | R | | | Bias | | |
|---|---|---|---|---|---|---|---|---|---|---|
| | | OL | ASS 4SG | ASS. 1SG | OL | ASS 4SG | ASS. 1SG | OL | ASS 4SG | ASS 1SG |
| Nov. 2005 | Ponte Felice | 0.9 | 0.951 | 0.95 | 0.978 | 0.985 | 0.985 | 1.038 | 1.007 | 1.002 |
| | Stimigliano | 0.617 | 0.92 | 0.62 | 0.955 | 0.969 | 0.956 | 0.978 | 0.997 | 0.978 |
| | Nazzano | 0.306 | 0.442 | 0.304 | 0.893 | 0.889 | 0.893 | 1.014 | 0.993 | 1.014 |
| | Ponte del Grillo | 0.482 | 0.688 | 0.479 | 0.895 | 0.926 | 0.894 | 1.022 | 1.004 | 1.023 |
| Nov. 2010 | Ponte Felice | 0.875 | 0.976 | 0.974 | 0.981 | 0.989 | 0.989 | 1.06 | 1.005 | 1.011 |
| | Stimigliano | 0.553 | 0.924 | 0.553 | 0.94 | 0.968 | 0.941 | 0.963 | 0.994 | 0.964 |
| | Nazzano | 0.674 | 0.9 | 0.672 | 0.884 | 0.952 | 0.885 | 1.036 | 0.994 | 1.036 |
| | Ponte del Grillo | 0.386 | 0.835 | 0.385 | 0.795 | 0.924 | 0.796 | 1.104 | 0.989 | 1.105 |
| Nov. 2012 | Ponte Felice | 0.912 | 0.958 | 0.958 | 0.971 | 0.979 | 0.979 | 1.007 | 0.997 | 0.994 |
| | Stimigliano | 0.85 | 0.94 | 0.869 | 0.965 | 0.974 | 0.968 | 0.953 | 0.993 | 0.957 |
| | Nazzano | 0.688 | 0.849 | 0.688 | 0.841 | 0.927 | 0.844 | 1.018 | 1.005 | 1.021 |
| | Ponte del Grillo | 0.641 | 0.832 | 0.638 | 0.826 | 0.925 | 0.829 | 1.037 | 1.016 | 1.046 |

**Table 1.** NSE, R and Bias for open-loop (OL), 4 stage gauges (A 4SG) and 1 stage gauge (A 1SG) data assimilation at Ponte Felice, Stimigliano. Nazzano, Ponte del Grillo stations for the 2005, 2010 and 2012 flood events

| Station | NSE | | R | | Bias | |
|---|---|---|---|---|---|---|
| | OBS | ASS SI | OBS | ASS SI | OBS | ASS SI |
| Ponte Felice | 0.912 | 0.913 | 0.971 | 0.971 | 1.007 | 1.009 |
| Stimigliano | 0.85 | 0.853 | 0.965 | 0.965 | 0.953 | 0.954 |
| Nazzano | 0.688 | 0.691 | 0.841 | 0.843 | 1.018 | 1.019 |
| Ponte del Grillo | 0.641 | 0.643 | 0.826 | 0.828 | 1.037 | 1.039 |

**Table 2.** NSE, R and Bias for open-loop and SI assimilation simulations at Ponte Felice, Stimigliano. Nazzano, Ponte del Grillo stations for the d 2012 flood event.

| Station | NSE | | R | | Bias | |
|---|---|---|---|---|---|---|
| | OBS | ASS TOT | OBS | ASS TOT | OBS | ASS TOT |
| Ponte Felice | 0.912 | 0.955 | 0.971 | 0.977 | 1.007 | 0.995 |
| Stimigliano | 0.85 | 0.936 | 0.965 | 0.973 | 0.953 | 0.987 |
| Nazzano | 0.688 | 0.839 | 0.841 | 0.924 | 1.018 | 1.01 |
| Ponte del Grillo | 0.641 | 0.825 | 0.826 | 0.922 | 1.037 | 1.019 |

**Table 3.** NSE, R and Bias for open-loop and both SG and SI assimilation (ASS TOT) simulations at Ponte Felice, Stimigliano. Nazzano, Ponte del Grillo station for the d 2012 flood event.

FIGURES

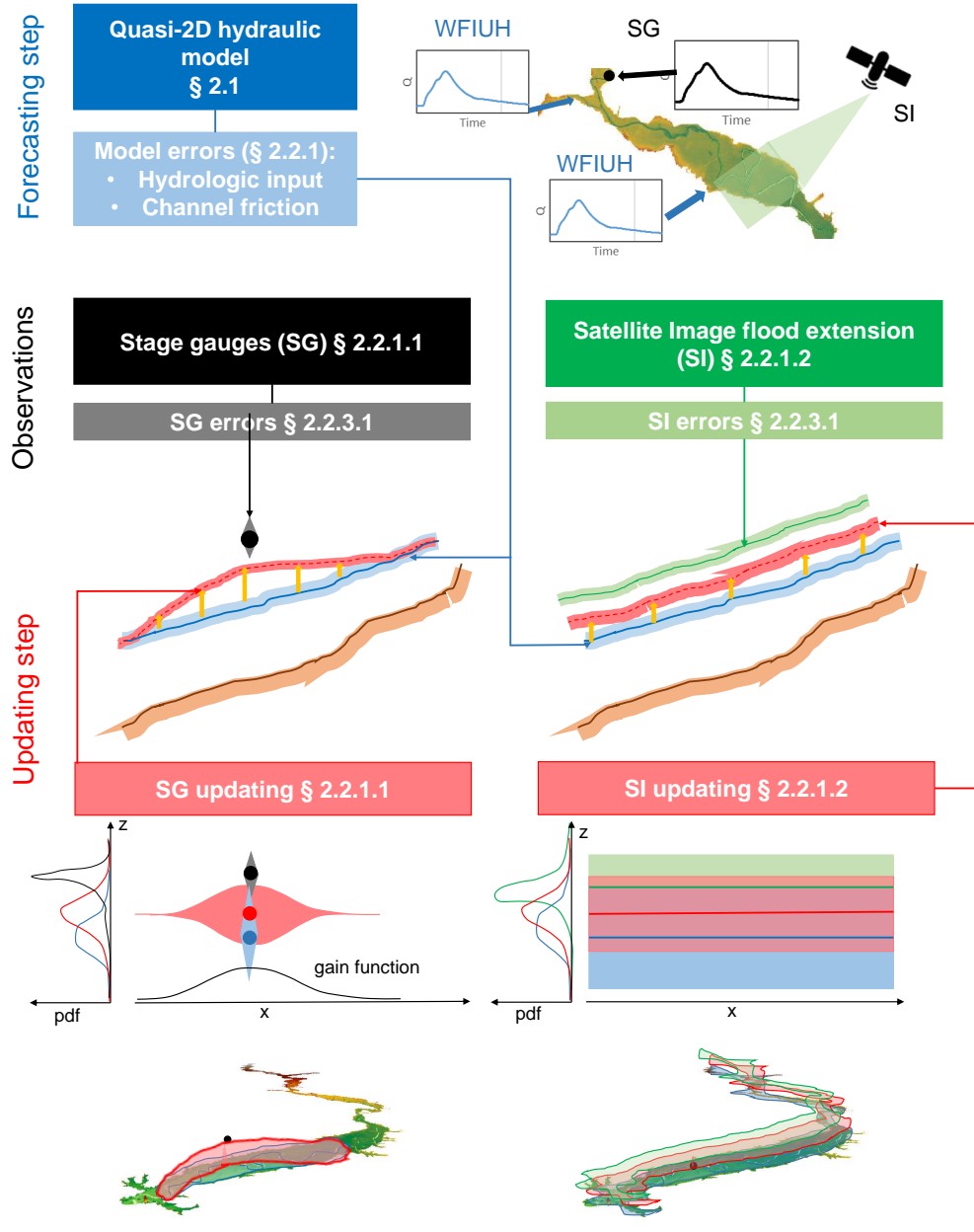

**Figure 1.** Scheme of the Data Assimilation(DA) framework

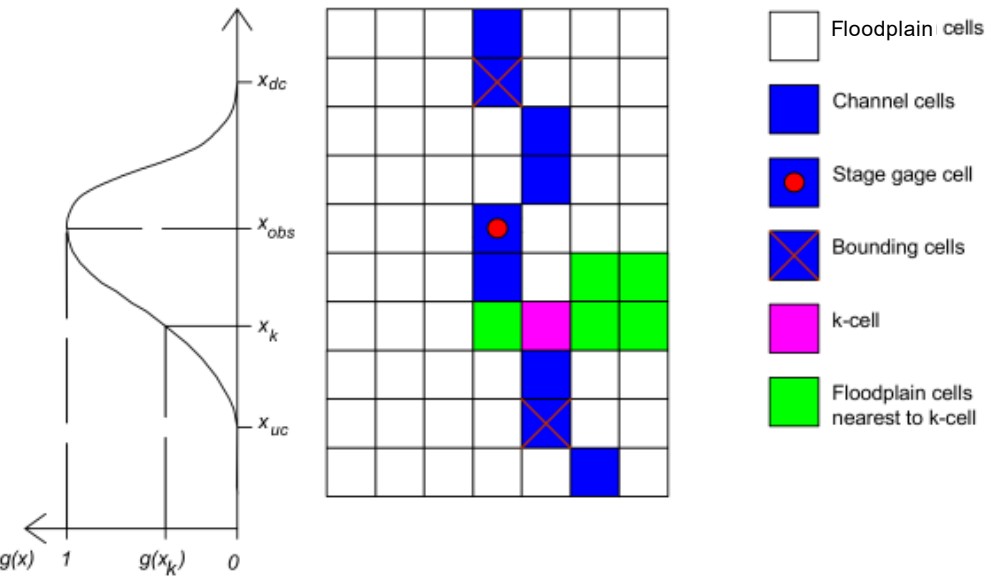

**Figure 2.** Scheme of the cells updating in the channel and floodplain domain adopting a gain function assimilating the stage gauges measurements

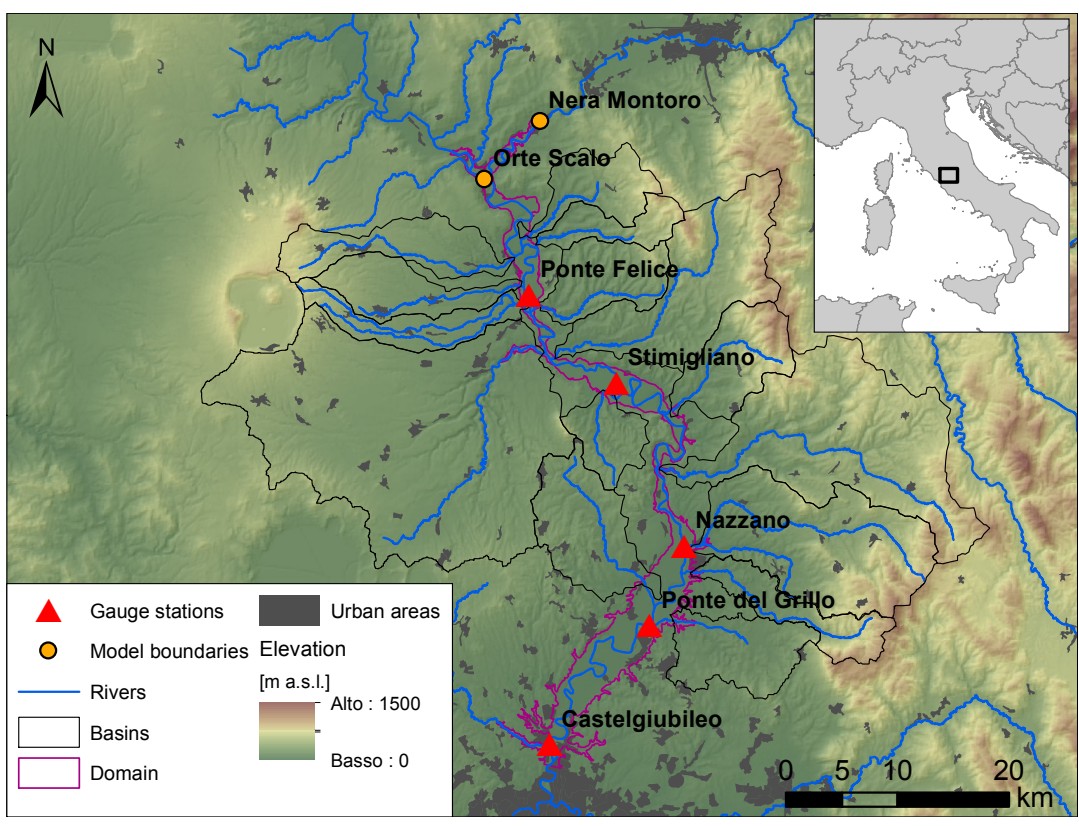

**Figure 3.** Map of the study basin with the contributing lateral river basins, the model boundaries and the reference gauge stations

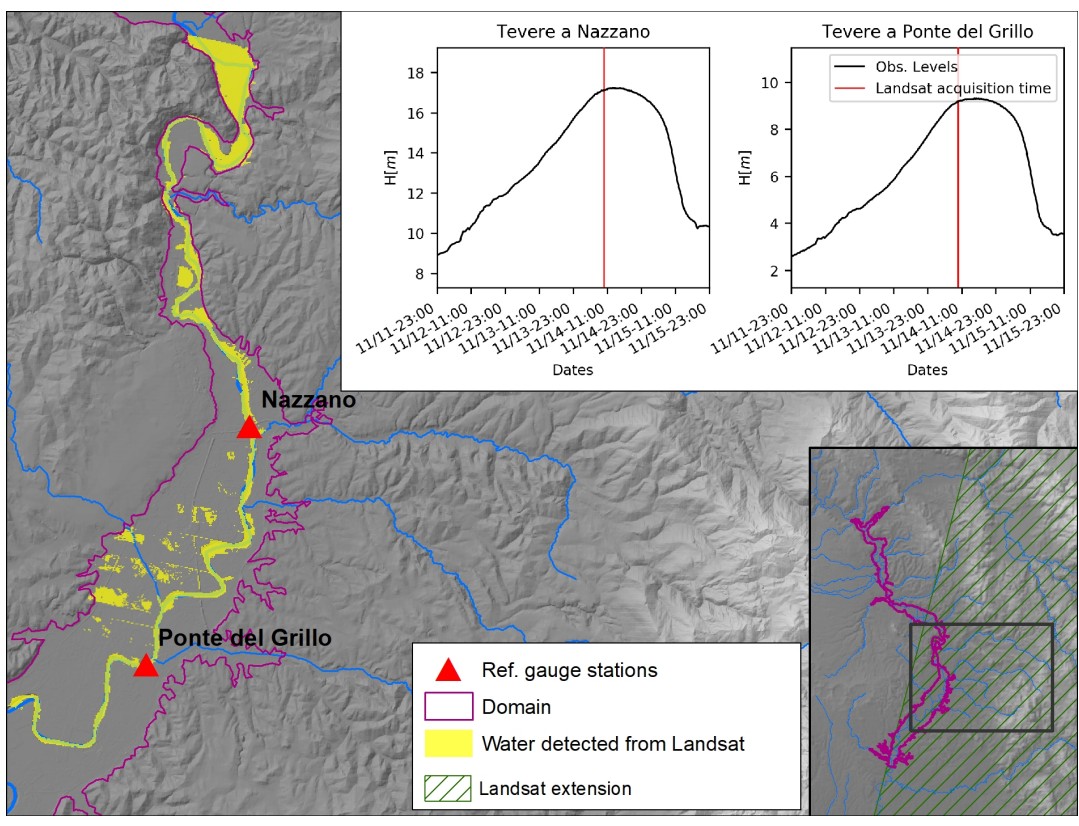

**Figure 4.** Extension of the water detected from the Landsat 7 image (acquisition date: 14/11/2012 - 09.43) in the computational hydraulic domain with the position of the Landsat acquisition time compared to the time series of the water depths in Nazzano and Ponte del Grillo gauge stations

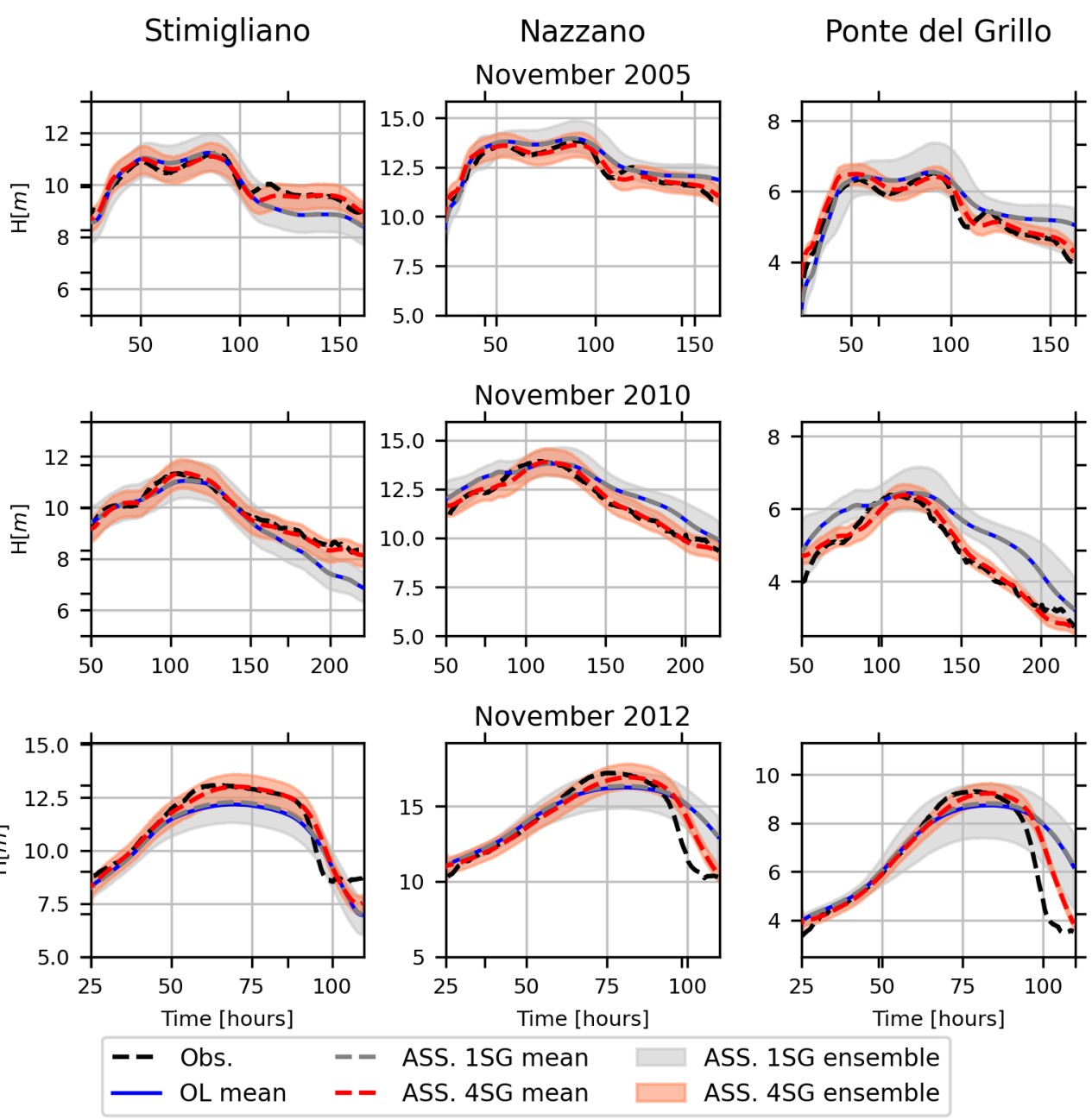

**Figure 5.** Water level time series at Stimigliano, Nazzano and Ponte del Grillo stations (respectively 16.6, 48.2 km and 59.1 km far from the upstream Ponte Felice station) for the 2005, 2010 and 2012 flood events: Observations (black, *Obs*), open-loop simulations (blue, *OL*), assimilation at 4 Stage gauges (red, ASS 4SG) and at the only upstream stage gauge (grey, ASS 1SG). Observations are assimilated every 15 minutes.

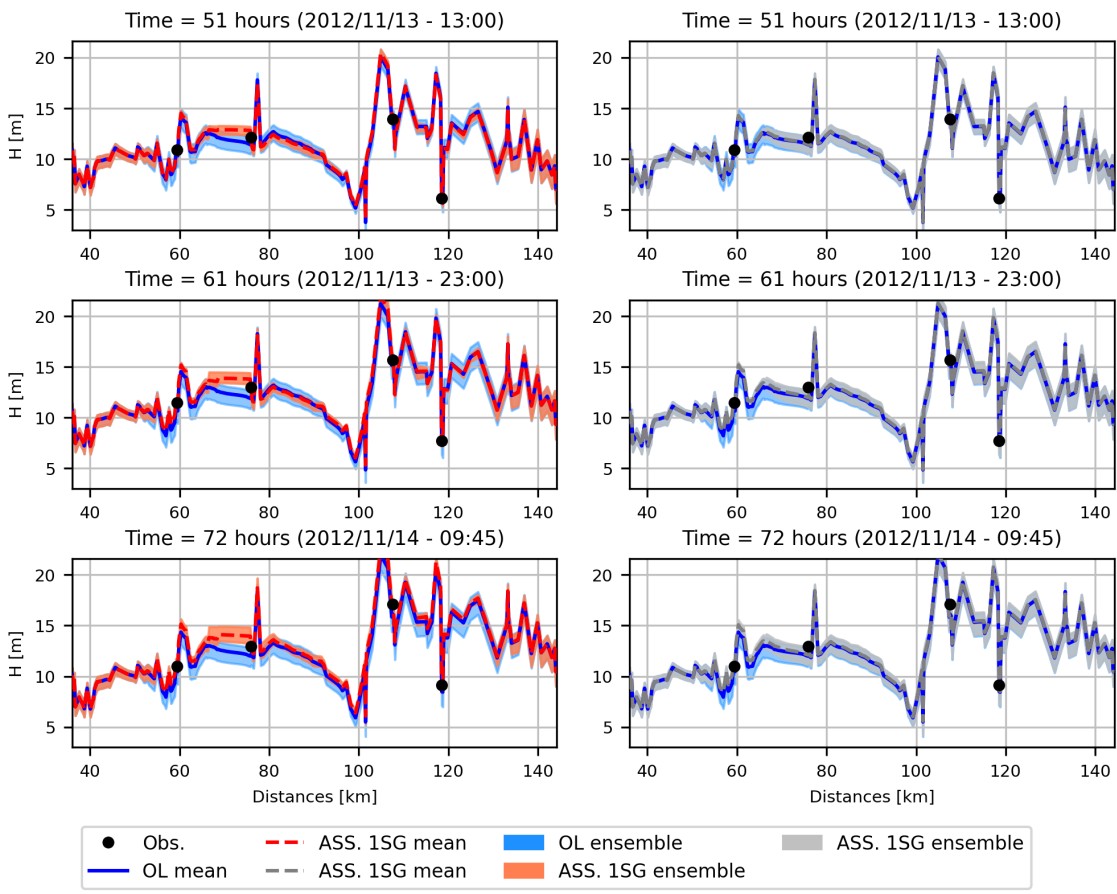

**Figure 6.** Plot of the channel water depths profiles for the open-loop simulations (blue, $OL$), assimilation at 4 Stage gauges (red, ASS 4SG) and at the only upstream stage gauge (grey, ASS 1SG) at three different time steps. Event: November 2012

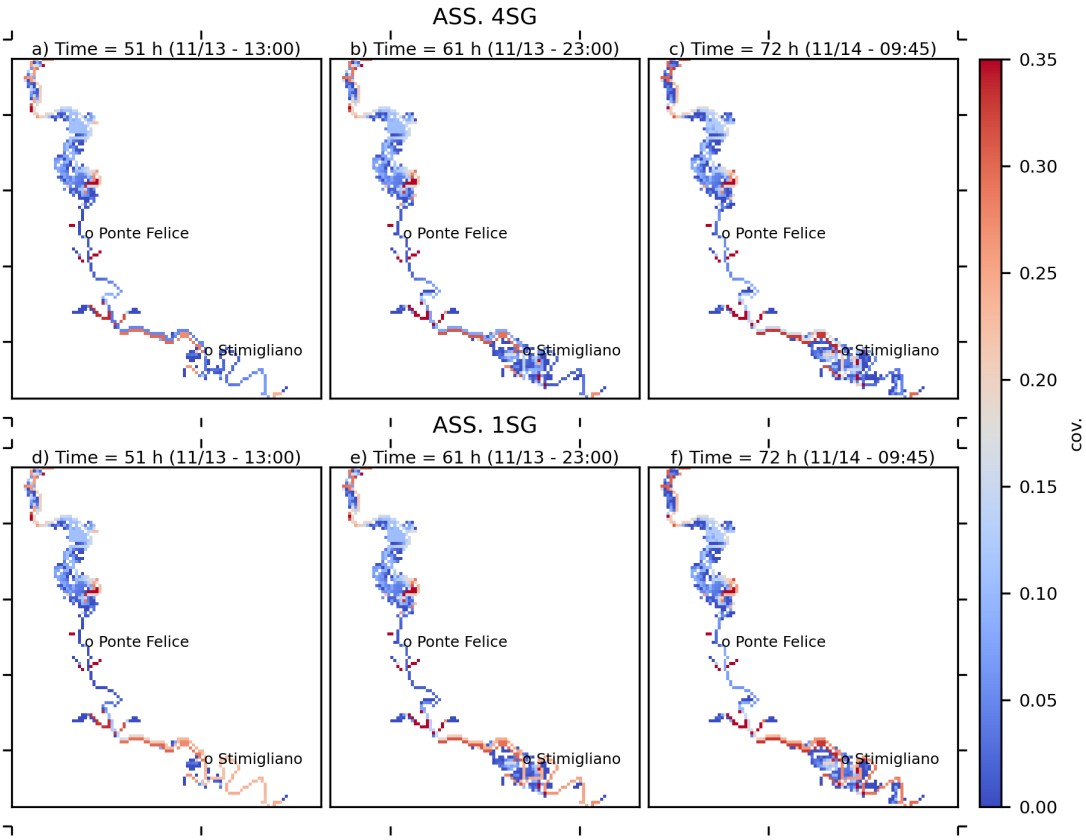

**Figure 7.** Map of spatial covariances for 4 Stage gauges - ASS 4SG- (a,b,c panels) and 1 stage gauge -ASS 1SG- (d,e,f panels) assimilation at three different time steps. Event: November 2012

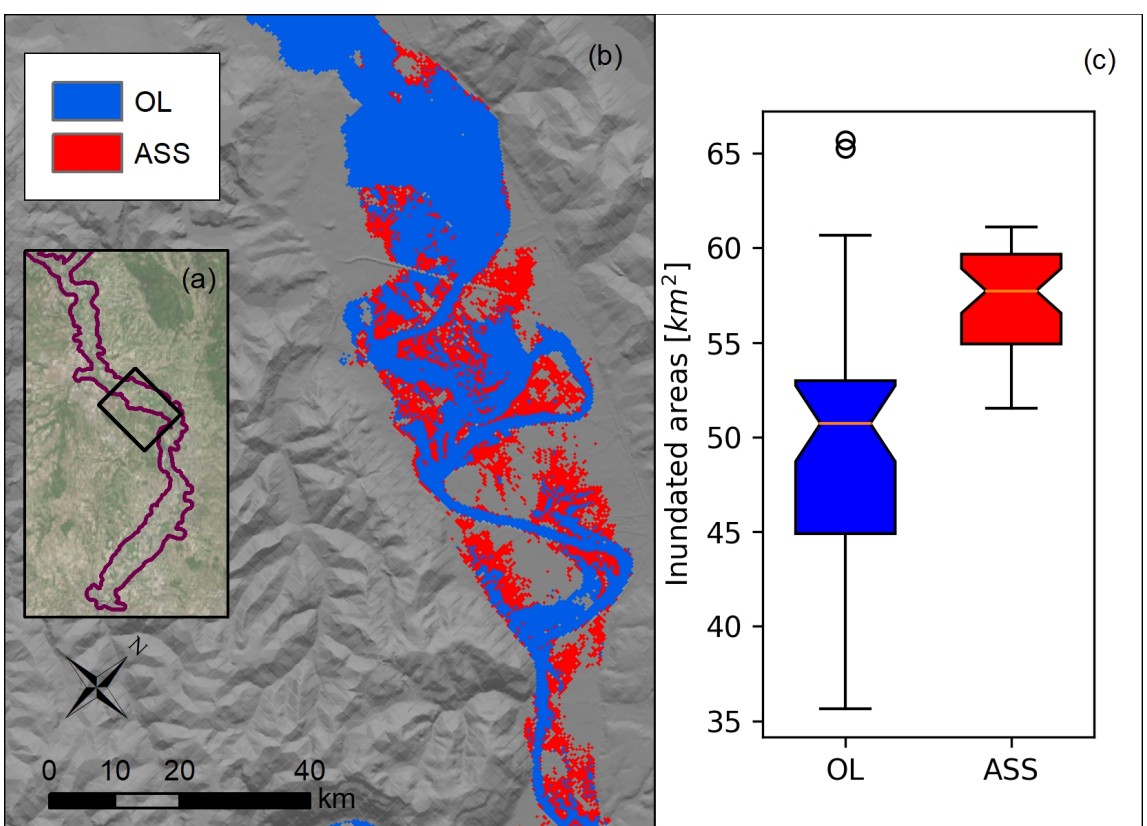

**Figure 8.** (a)Location of Figure 8(b) area compared to the extension of the computational domain (in purple). (b) Flood extension related to the average water levels for open-loop (OL) and stage observations assimilation (ASS 4SG) at the time of the Satellite Image acquisition. Note that OL flood extension includes also ASS 4SG flood extension. (c) Boxplot of the flood extension considering each element of the two ensembles. Event: November 2012

.

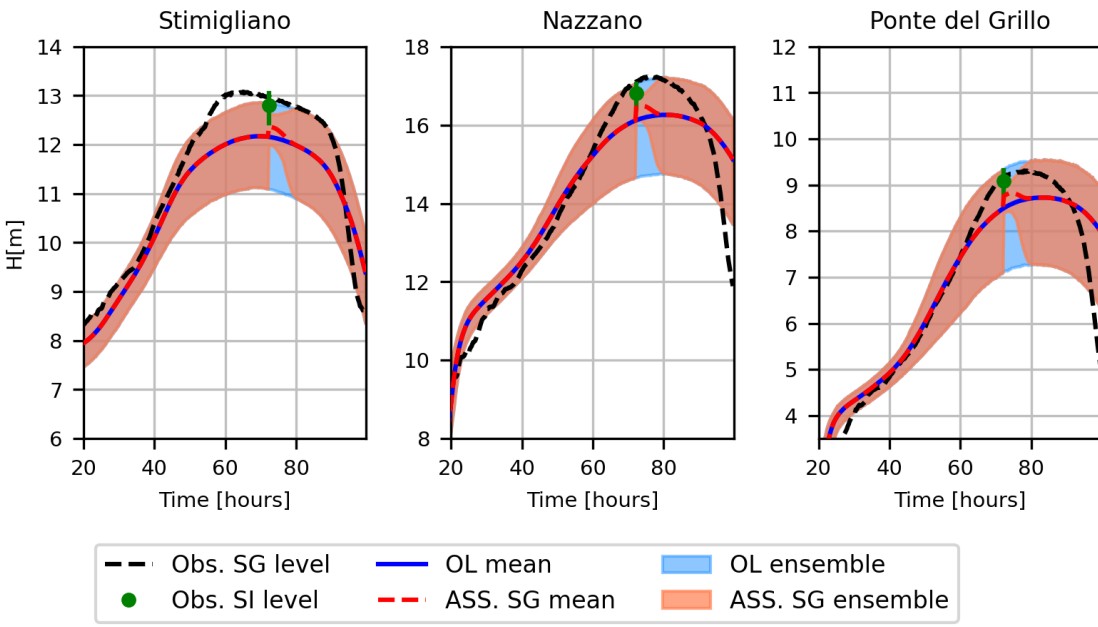

**Figure 9.** Water level time series at Stimigliano, Nazzano and Ponte del Grillo stations for the 2012 flood event: Observations (black), open-loop simulations (blue), and assimilation of the indirect observations from the Satellite image (red).

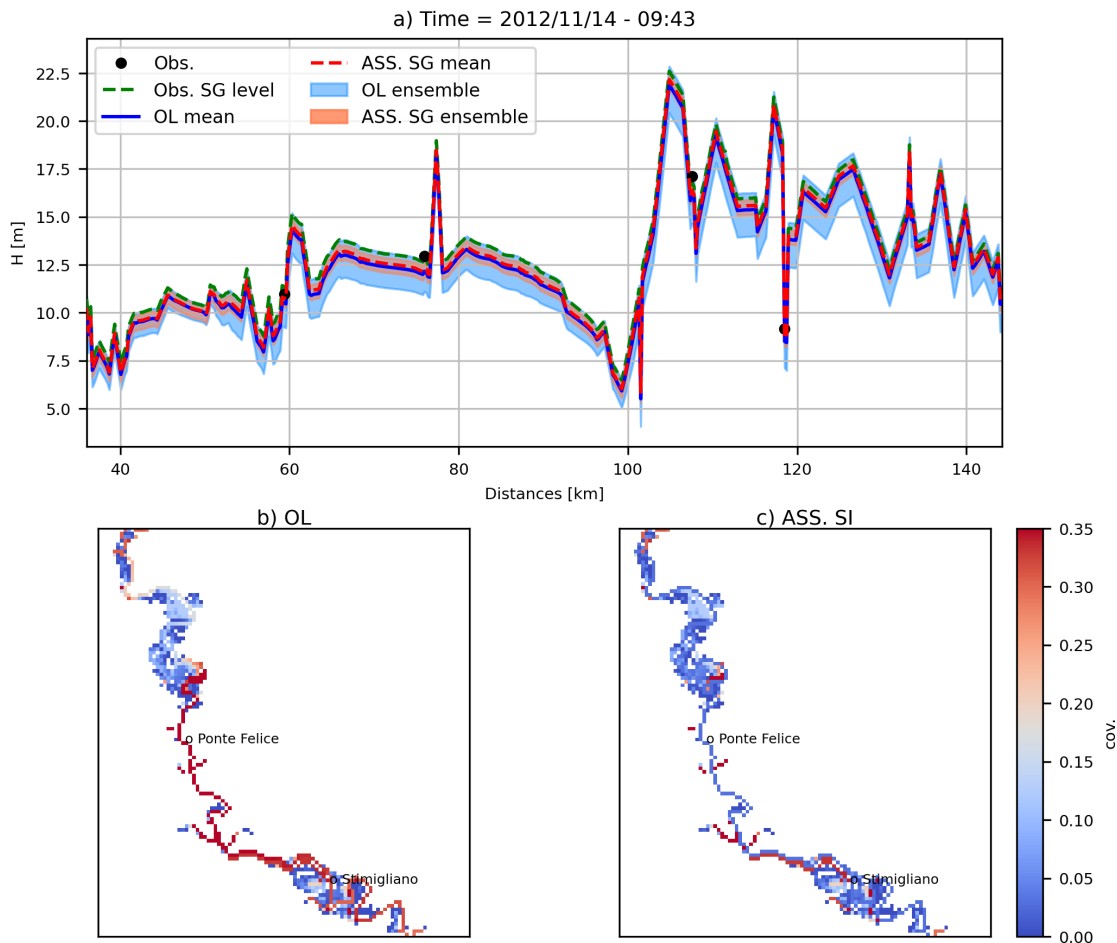

**Figure 10.** a) Plot of the channel water depth profile for the Open-Loop (OL) and Satellite Image data assimilation (ASS SI) simulations at the SI acquisition time.b) and c) are the maps of covariances at the SI acquisition time of the OL and ASS SI simulations. Event: November 2012

.

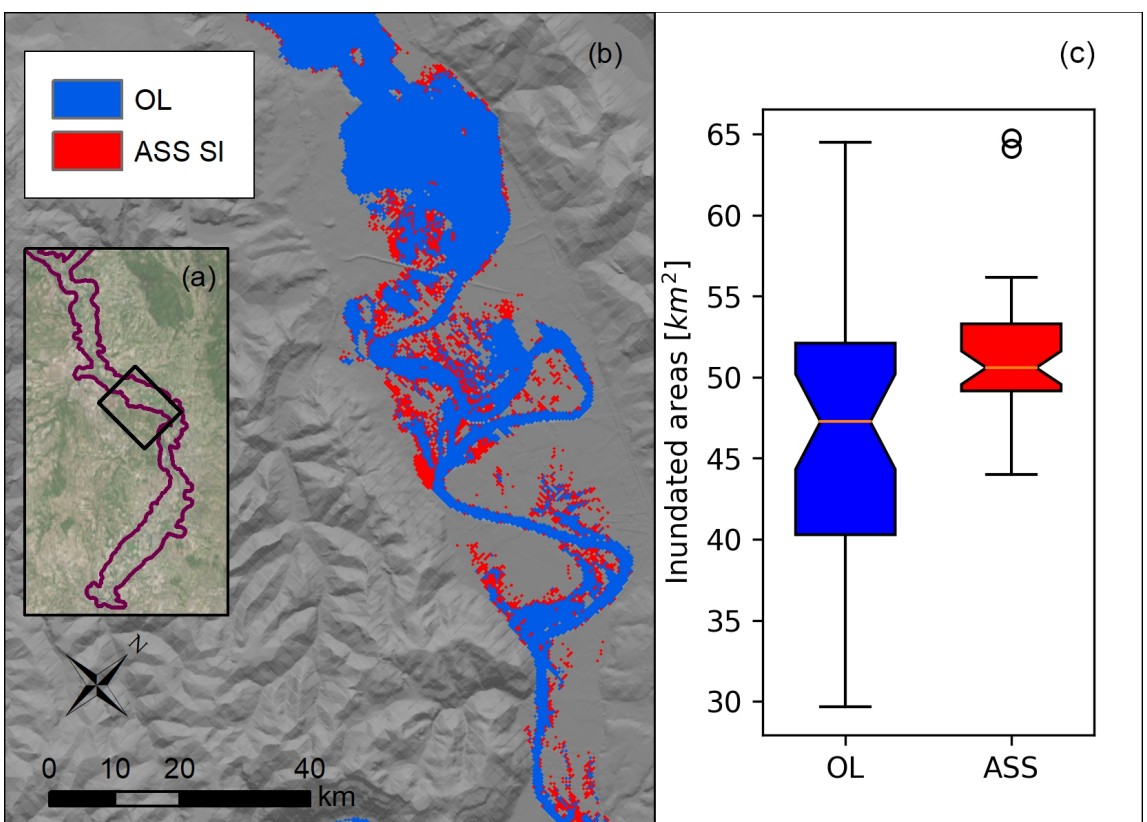

**Figure 11.** (a)Location of Figure 11(b) area compared to the extension of the computational domain (in purple). (b) Flood extension related to the average water levels for open-loop (OL) and assimilation of the satellite image (ASS SI) at the time of the Satellite Image acquisition. Note that OL flood extension includes also ASS SI flood extension. (c) Boxplot of the flood extension considering each element of the two ensembles. Event: November 2012

.

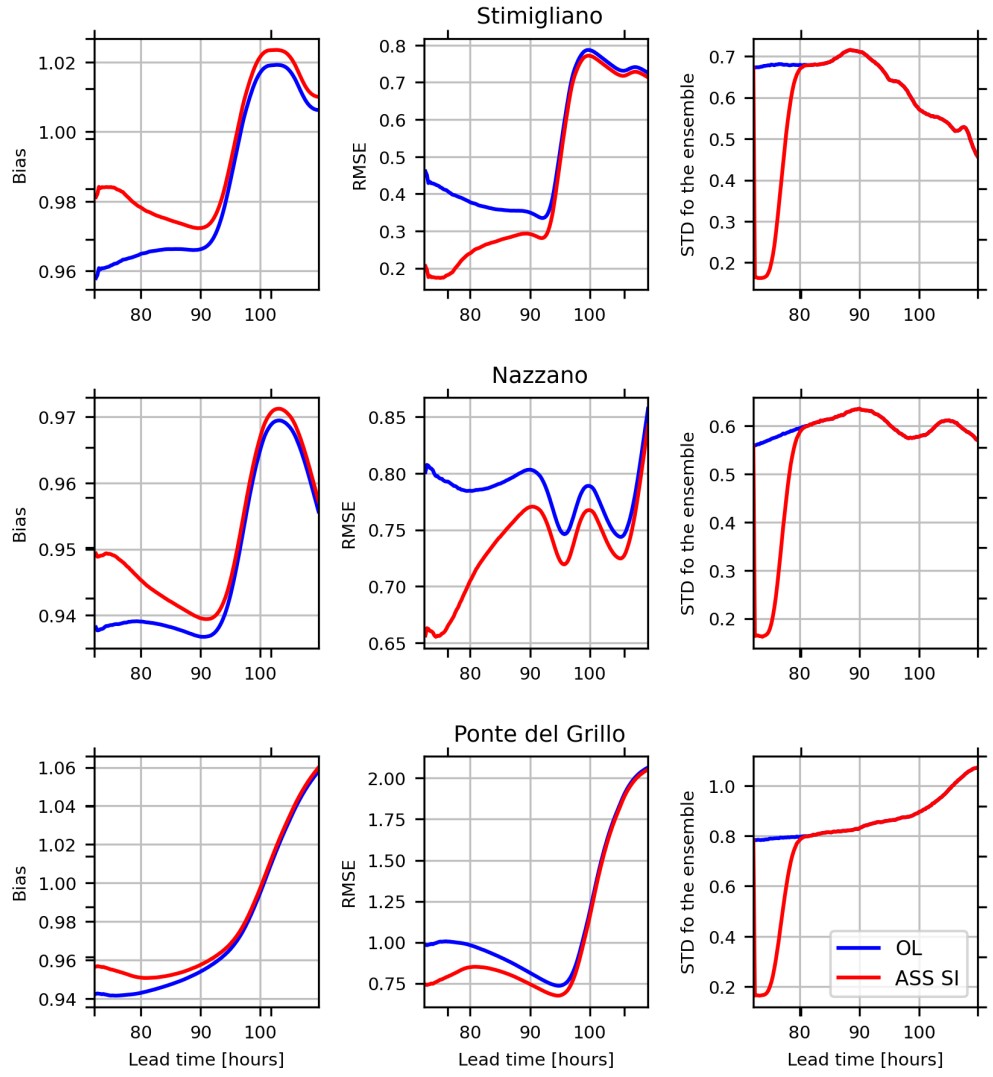

**Figure 12.** Performance indexes (Bias, RMSE and variance of the ensemble spread) with the lead time after the acquisition time of the SI observation at Stimigliano, Nazzano and Ponte del Grillo stations. Event: November 2012

.

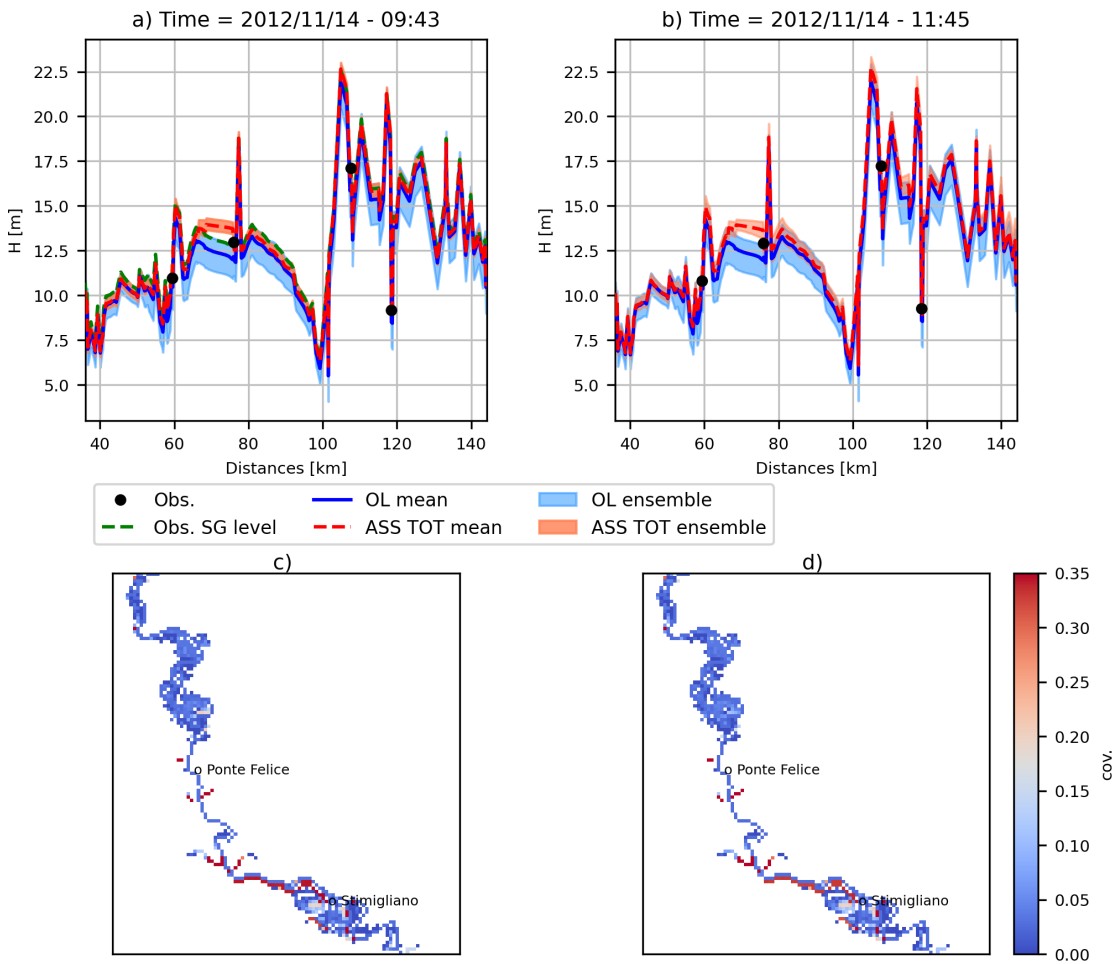

**Figure 13.** Channel water depth profiles ad spatial covariances at the time of the SI acquisition (left panels) and 2 hours later (right panels)
.

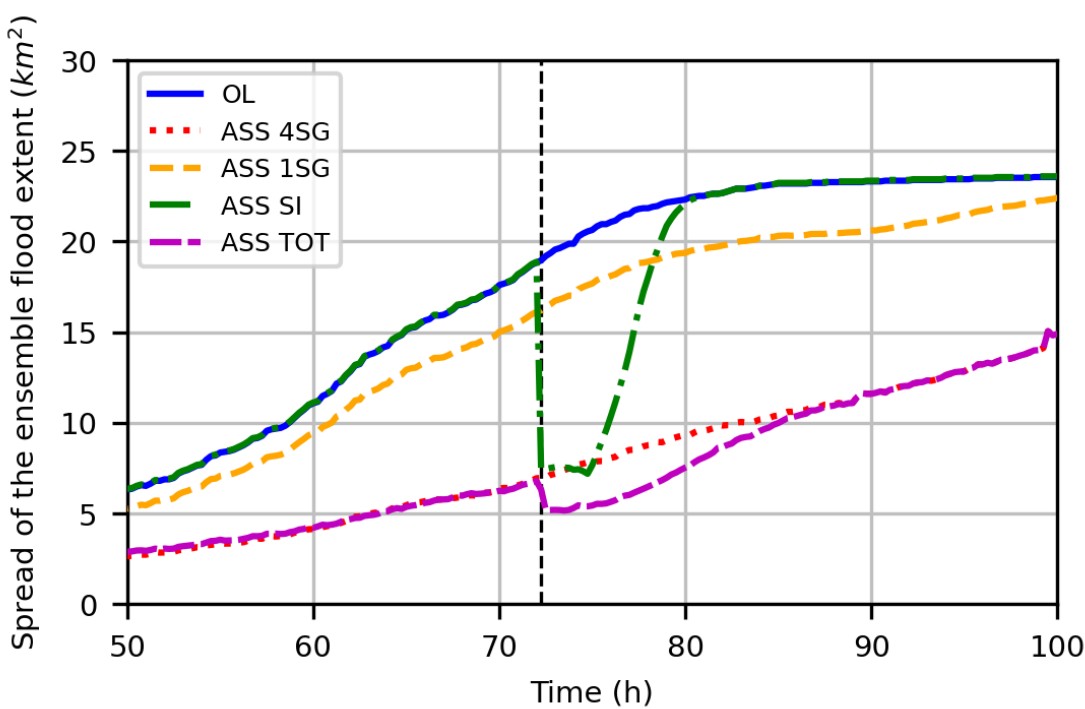

**Figure 14.** Maximum flood extent variability versus time for Open Loop (OL), assimilation of four (ASS 4SG) and one (ASS 1SG) stage gauge(s), SI assimilation (ASS SI) and both 4SG and SI assimilation (ASS TOT). The vertical black dashed line indicates the time of the SI acquisition. Event: November 2012

.