# Peer review of "Simultaneous assimilation of water levels from river gauges and satellite flood maps for near-real time flood mapping"

_Hydrology and Earth System Sciences, 2021_

## Author Response (AR1)

**Response to Referees' Comments**

**1) Referee #1**

**ID: R1_01**
**Referee comments**:
General comments:
This manuscript shows a case study with assimilation of real observations of satellite-derived flooded areas and local gauge observations to keep a flood forecast model on track. The paper is well written and the topic is relevant for HESS.
The authors use the (perturbed-observations) EnKF to update the state of a 2D hydraulic model, coupled to a hydrologic model, which provides the inflows as input the the hydraulic model domain. The hydrologic component and global model setup is fine from this Referee's point of view, and the authors have done a good effort throughout the study.
However, after reading the manuscript, one is left with the impression that on the assimilation approach the authors have come with a "preconceived" plan without seriously considering standard assimilation possibilities that are already out there. It is my belief that studies that defend alternatives methods should always consider existing approaches for benchmarking. Otherwise, there is the risk that we are left with a bunch of alternatives and without enough information to decide whether it is worth (or better) to try one or another. Also, one is left with the impression that the overall results in this study could be rather different using alternative assimilation approaches. This does not invalidate the study at all, but it should be augmented with results from applying (assimilation) techniques that are already on the table.
The joint assimilation of observations from satellite and local river gauge time-series is of clear interest in this context, and it is natural that flood forecast feeds on as many observation types as possible. However, despite the effort put by the authors, the above issue plus a scarcity in the section for results is, in my opinion, a missing opportunity.
**Authors' reply:** We thank the Referee for the solid review that helped improving the manuscript and better elaborate some parts of the Data Assimilation application and the results. We better clarified the aim of the work, pointing out that the main novel part is the joint assimilation of both traditional static sensors and satellite derived flood extent for improving the performances of a distributed unsteady bidimensional hydraulic model. The use of Quasi-2D hydraulic modelling is a challenge for real time DA frameworks and injecting distributed EO data (in addiction to single point static sensor observations) makes the challenger harder. To our knowledge, while there are a number of papers also using 2D hydraulic modelling (see ID: R1_06 comment) this is a quite challenging and still a much debated topic requiring further research to develop methods, procedure towards DA that include 2D models. Moreover, we also better specified why using some novel approaches for updating the state variables in the DA framework.
**Authors'actions:** In the following lines we isolated every Referee comment assigning a specific ID with a progressive number (e.g. R1_XX) and our point-by-point reply.

**ID: R1_02**
**Referee comments**:
In the introduction, it would have been also nice also to refer, somehow, to the advances in Numerical Weather Prediction (NWP) to frame the context of the early warning systems.
**Authors' reply:** We agree with the Referee' suggestion.
**Actions:** We added some references on the NWP in the introduction (lines 34-88):
*"In case of medium term forecast (i.e. days/weeks ahead), rainfall and runoff observations are not sufficient and Numerical WeatherPrediction (NWP) models are required, especially for basins whose concentration*

*time is limited so that emergency measures, such as evacuation, cannot be properly applied on time (Hopson and Webster, 2010). In this regard, recent advances in NWP models in weather forecasting were developed adopting ensemble prediction systems (EPS) (Buizza et al., 2005) as inputs of hydrological and hydraulic models."*

And (lines 47-52):

*"DA models are used both in NWP and hydrologic-hydraulic modelling. Advances in EPS approaches and increasing of computational power allowed to improve the accuracy of NWP models as inputs of flood forecasting systems (Yu et al., 2016). Successful examples of advanced EPS approaches in NWP models for flood forecasting services at large scale are the EPS-ECMWF - from the European Centre for Medium Range Weather Forecasts- (De Roo et al., 2003) and the COSMO-LEPS - from the Consortium for Small-Scale Modelling – Limited-area Ensemble Prediction System (Marsigli et al., 2005). Flood models can be updated in DA approaches by ingesting outputs of NWP models or direct rainfall-runoff observations."*

**ID: R1_03**

**Referee comments**:

It is not easy to select a starting point as best reference on assimilation in this field along the chain of available papers, ranging from general methodological assimilation papers (outside from hydrology) to more applied ones in this specific field of flood forecast. Perhaps, on the assimilation of real satellite-based flood extent observations with a 2D model stretching over a number of rivers (some main rivers plus some tributaries), and using an upstream coupled hydrologic model for the inflow timeseries (as well as estimating online model parameters, as this study), the authors should look back to García-Pintado et al. (2015). From there, they could also further look into citing articles to go forward in time towards more recent studies.

**Authors' reply:** We thank the Referee for the suggestion.

**Actions**: In the introduction, we extended the analysis about the state of the art of the scientific literature on Data Assimilation applications with hydraulic modelling for flood forecasting including also García-Pintado et al. (2015) reference. We report here an extract of the introduction (lines 60-109):

[revised manuscript text omitted]

**ID: R1_04**
**Referee comments**:
For a more general context about flood forecast considering assimilation, it is also advisable to read the review by Grimaldi et al. (2016). Although this paper is cited here, the manuscript indicates the authors have not actually gone through the paper details.

**Authors' reply:** We agree that Grimaldi et al., (2016) provided a good starting point as framework on the state of the art of DA modelling for flood forecasting.
**Actions**: According to the referee suggestions, we extended the introduction considering Grimaldi et al. (2016) (See lines 62-85, 88-99)

**ID: R1_05**
**Referee comments**:

Last but not least, it is surprising that despite the effort put into this study the authors do not show any spatial graphical output (despite they use a 2D model) for diagnosis of the assimilation. They could well show maps with, for example, assimilation increments at specific times, covariances, etc. These kind of 2D plots are very helpful to try to understand what is under the hood in the assimilation. This greatly eases the understanding and evaluation of what is working well or not.

**Authors' reply:** We agree with the referee concern. This suggestion represents a quite significant improvement of the revised work.

**Actions**: We added new Figures showing the distribution of the covariances for different time steps for the 2012 flood event in case of SG and SI assimilation (Figures 7, 10, 13)

**ID: R1_06**
**Referee comments**:
Specific comments:
L45. Some of these studies do use a 2D hydrodynamic model. In summary:
Andreadis et al., 2007  : 2D
Matgen et al., 2007:      1D
Neal et al., 2009;        1D
Hostache et al., 2010:   2D
Matgen et al., 2010:      1D
Giustarini et al., 2011:   1D
García-Pintado et al., 2013: 2D
Mason et al., 2012: no DA but aimed to 2D
Andreadis and Schumann, 2014:  2D
What seems to indicate that the authors have actually not read these papers.

**Authors' reply:** We thank the Referee for pointing out this error.

**Actions:** We corrected the sentence as follows (lines 60-63): "*To tackle these issues, in the last ten years, Earth Observation (EO) data were used to inject water altimetry observation in DA frameworks for updating flood models, usually adopting radar Synthetic Aperture Radar (SAR) technologies and 1D (Matgen et al., 2007; Neal et al., 2009; Matgen et al., 2010; Giustarini et al., 2011) or 2D ( Andreadis et al., 2007;Hostache et al.,2010; Mason et al., 2012; García-Pintado et al., 2013; Andreadis and Schumann, 2014) hydraulic routing algorithms.*"

**ID: R1_07**
**Referee comments**:
L57: Clarify here in which sense is the DA framework novel? Specifically, the EnKF has now a long history.

**Authors' reply:** We agree that the novelty of the proposed research is not the DA model itself (EnKF is a consolidated approach).

**Actions:**  we removed the "novel" word for the DA framework and we better clarified the novel aspects of the proposed research (lines 109-118):

"*Despite the remarkable progress in the integration of remotely sensed observations in DA frameworks, there are still majorchallenges that need to be faced (Grimaldi et al., 2016). For example, there is not still in scientific literature an approach able to assimilate heterogeneous observations from both local and distributed datasets coming from different sources (i.e. traditionalstage gauges and remotely sensed flood extents). Moreover, Quasi-2D and 2D hydraulic models can be sensitive to differentsimultaneous local state updating (i.e. water level corrections at specific time steps), because contiguous channel/floodplaincells  can be  characterized  by  different  elevations,  geometry  and  roughness,  therefore  instability  issues  can  rise during  themodel corrections. Another critical issue is that large scale flood forecasting models need to provide timely predictions but their spatial resolution can limit the effectiveness of the assimilation of satellite derived flood extents (Hostache et al., 2018).In this work, a DA framework supported by*

*heterogeneous observations coming from both local water level observations(i.e. stage gauges)  and spatially distributed information gathered from satellite images - is proposed and tested. This researchseeks to develop a more flexible DA scheme that may value all available sources of observations for distributed flood modellingupdates. The aim of this work is to mitigate flood prediction uncertainties by combining heterogeneous data and an integrated topographic-hydrologic-hydraulic modelling approach, while maintaining inundation forecasting robustness, scalability andnumerical stability. In achieving this goal, novel scientific advances and technical challenges of EO-driven DA approaches forflood prediction are investigated and in particular: A methodology for updating the state variable from multiple local stagegauges observations of a hydraulic model for distributed flood routing in floodplain domains; the gathering of spatially dis-tributed water level observations by means of flood extension processing and detection from satellite images, also adopting GIS algorithms for overcoming the issues of the different resolutions between the ensembles of the flood extents retrieved fromthe satellite derived images and the ones generated from the hydraulic model simulations. "*

**ID: R1_08**
**Referee comments**:
L77: The regular grid and simple IO formats do not make the model more "suitable" to DA than using an unstructured mesh or more complex (e.g. hierarchical) IO formats. This just allows for a simpler code.
**Authors' reply:** we agree with the Referee comment.
**Actions:**  We replaced "suitable" with "simpler"  (line 143)

**ID: R1_09**
**Referee comments**:
L94 "assess maximum flood energy gradients". How is this relevant? How is energy coupled with the 2D flood model or used here? It is also unclear if the floodplain computational domain evolves with time along with model integrations or is preset, based on GFPLAIN.
**Authors' reply:** We agree that the description of how we defined the computational domain (that however does not evolve with time) can move the reader's attention away from the main purpose of the manuscript and it is not relevant to this study.
**Actions:**  We removed this paragraph.

**ID: R1_10**
**Referee comments**:
L114: No uncertainty is taken into account in the rainfall input. It is worth to a) discuss briefly the errors in rain gauge data [e.g. the possibility of generating quantitative precipitation ensembles via Sequential Gaussian Simulations, etc.] and how the uncertainty is propagated downstream in the forecasting chain, and b] some reference to coupling with [possibly ensemble] NWPs.

**Authors' reply:** We agree that we did not directly take in to account the uncertainty in rainfall input. However, we expressed the inflow uncertainty including  all the sources of uncertainty of the hydrologic modelling. In this revied version, we included the temporal correlation of the inflow errors and the standard deviation of the white noise component is derived considering  the frequency distribution of its relative flow errors (observed versus simulated flow values) obtained by the calibration and validation of the hydrologic model.
We calibrated and validated the hydrologic model considering  four small ungauged basins of the Tiber river basin in order to find the optimal values of the channel/hillslope flow velocity and infiltration parameters.
Since we directly compared the simulated and the observed streamflow values, the standard deviation error equal to 0.28, indirectly takes in to account all the sources of errors: from the rainfall

inputs and its spatial distribution along the basin, the simplified modelling of the flow routing, the neglected physical processes, such as the groundwater flow, the mud and debris flow, the antecedent soil moisture conditions.

We agree that more refined methods such as the generation of quantitative precipitation ensembles should be mentioned.

**Actions:** we added the following lines (449-458): "*The hydrologic model is affected by different sources of uncertainties: the structural uncertainties, given by the simplification of the modelled physical processes (e.g. we adopted a simplified lumped WFIUH approach, neglecting groundwater flow,mud and debris flow), the input uncertainties (given by the rainfall values and antecedent soil moisture conditions), and parametric uncertainties due to the inaccuracy of the model calibration). All of these sources of uncertainty should be considered. For example input rainfall uncertainty from rain gauges can be estimated considering quantitative precipitation ensembles (Clark and Slater, 2006), such as Sequential Gaussian Simulations (Goovaerts et al., 1997; Rakovec et al., 2012a). Precipitation ensemble generated with NWPs can than be coupled with hydrologic models to improve flood forecasting (Jaspe ret al., 2002; Sorooshian et al., 2008). In this work we decided adopted a simplified procedure taking in to account all the modeling uncertainties considering the frequency distribution of the errors between the observed and simulated flow values obtained by the calibration and validation of four small tributaries of the Tiber river basin in past flood events.* "

**ID: R1_11**
**Referee comments**:
Section 2.2
L140: Satellite and river gauges give observations as water levels. It appear as more natural to use the forward operator H to map the model state into the the observation space.
**Authors' reply:** we used the H as identity matrix to map the model state into the observation state. See the following lines (204-205): "
*For this reason, the observation transition operation H introduced in Eq. (2) is an identity matrix, being a direct relationship between state variables and observations.*"

**ID: R1_12**
**Referee comments**:
Section 2.2.2
L155: "significant". Please reserve the term "significant" for its statistical meaning in manuscripts. This is just a threshold. Use simply "high".
**Authors' reply:** we agree with the referee's suggestion
**Actions:** We replaced "significant" with "high" (line 232).

**ID: R1_13**
**Referee comments**:
Overall, I'm sorry to be so negative here. Why, not just use covariances for the simultaneous assimilation of several observations. This whole section [specifically weight the observation values based on inverse distance weight] goes against the whole spirit of the EnKF. If the authors believe their approach is better than using covariances to control the updating steps (as standard), they should at least put this into context to defend their approach.
The authors should look at approaches that have been deeply studied (as localisation and ensemble inflation), and refer to them. There is plenty of bibliography (mostly on the NWP field) on this. It is even better if they can actually implement an EnKF with localisation (and, possibly, inflation). Then use this as benchmark to evaluate if their approach actually makes things better/worse/different...

**Authors' reply and actions:** We thank the Referee for the comment that helped clarifying the reasons why we adopted our methodology (alternative to localization) that goes against the spirit of EnKF.

In our model, we could use covariance localization considering absolute water levels (as respect to the average sea level) or water depths (as respect to the terrain elevation)

However, river water depths can dramatically change among contiguous cells of the hydraulic domain for example moving from a channel and a floodplain cell or because changes of the local geometry (e.g. cross section shape). In fact, usually stage gage measurements are located under hydraulic structures such as bridges, where the geometry of the cross section (that can be reshaped to be adapted to the bridge geometry) can have high differences as respect to the surrounding natural cross sections. Therefore, the standard localization techniques could give some issues (e.g. numerical surging or filter convergence) when updating a state variable close to the observation location but considerably different to it.

Therefore, following the Referee's advice, we applied an observation localization technique considering absolute water levels. Specifically, we implemented the localization methodology applied by Garcia-Pintado et al., 2015, as rare case study of localization application in a 2D hydraulic model. They adopted an Observation Localization method applying a weight to the error covariance that is a distance metric based also on a channel network distance. The along network distance is more physically meaningful as respect to the Euclidean distance.

However we encountered instability issues when updating the water levels far from the observation location especially in those areas with higher channel slope because the changes of terrain elevation from upstream to downstream. This instability issue motivated us to not apply a standard localization technique, but propagate the correction (the difference between the posterior and the forecast state) upstream and downstream with our proposed methodology, that.

We updated the manuscript text illustrating the state of the art and the issues of localization in the methodology.

Moreover, we also implemented the inflation as suggested by the reviewer.

**ID: R1_14**
**Referee comments**:
Further, to stop the model each time a new observation arrives, as it is described, seems rather ineffective regarding the assimilation of the river gauge continuous time-series. The authors could look at asynchronous assimilation approaches, so it is not strictly needed to stop the model as frequently.

**Authors' reply:** We thank the Referee for the good suggestion. The need of stopping the model and saving the binary files is just a technical reason that can be solved in a future step accessing to the hydraulic machine code .

Asynchronous assimilation approaches can be a valid alternative to reduce the number of model stops, even if each time step they are used, they require little higher cost to the computational time and the storage requirements.

**Actions:** We mentioned the potential adoption of asynchronous approaches as future investigation in Section 4.4 as follows (lines 612-618): "*The need of updating the model states each time the stage gauge observations are available can affect the efficiency of the model in terms of computational performance. Alternative approaches such as the Asynchronous Ensemble Kalman filter (AEnKF, Sakov et al., 2010), allowing to ingest past observations over a time window, to update model state at a specific time step, could help reducing the times when the model is stopped, even if each model updating require little higher costs to the computational time and higher storage requirements*"

**ID: R1_15**
**Referee comments**:
Section 2.2.3
L205: provide some citation (i.e. to George Matheron) for Kriging.
**Authors reply and actions:** We added some citations *"(Matheron,1969; Oliver and Webster, 1990)" (*line 291)

**ID: R1_16**
**Referee comments**:
Section 2.2.4 Model errors
L222 The $Q^{true}$ should be $Q^{observed}$. Also, the inflow error is simulated as as white noise. This is hardly realistic and, beyond this, a more realistic noise should actually improve the assimilation. I believe this partly explains the lack of success shown by Figure 8.
**Authors' reply and actions:** $Q^{true}$ can be the Q indirectly observed from the stage gauges or simulated by the hydrologic model.  We changed this as  $Q^{os}$ specifying the this is the observed (derived from the flow rating curves) or simulated inflow.
We also agree that the white noise is not realistic for flows derived both from stage gages ($Q_{SG}$) and from hydrologic modelling ($Q_I$).
To take into account the temporal correlations of the inflows, we modified the inflow perturbation. Following the procedure proposed by Garca-Pintado et al. (2013), we considered the matrix of inflow errors for a generic time step *t* as the element-wise product between a emporal correlated errors (following Evensen, 2003 approach) and the heteroscedastic error, whose variance is proportional to the flow value at time t.
We didn't consider a spatial correlation for the SG-derived observations because errors in stage measurements and uncertainties in rating curves are normally independent between sites. On the other hand, we considered a spatial correlation in the inflows derived from the hydrologic model considering a Gaussian-decay correlation model based on the distance between the locations of the point inflow boundary conditions (Garcia-Pintado et al., 2013).
Finally, we believe that the "lack of success" (i.e. the limited persistence of the model performances) is due to the sole model state correction without updating the model inputs. This limited persistence of the model is confirmed by other studies (Andreadis et al., 2007; Matgen et al., 2010; García-Pintado et al., 2013; Andreadis and Schumann, 2014).

**ID: R1_17**
**Referee comments**:
Section 2.2.5
L242    2.2.6 should be 2.2.5.1
**Author's reply and actions:**  Thank you, there were some issues at the fourth level of the subsection formatting. We corrected all the sub sections' numbers.

**ID: R1_18**
**Referee comments**:
L248   Again, replace "true" by "observed". And, again, hard to believe the lack of temporal correlation in these errors.
**Authors' reply an actions:** We agree on replacing the "true" superscript. We replaced $h^{true}$ with $h^{obs}$. The temporal correlation was added to the stage gages observations.
**Actions:**

**ID: R1_19**
**Referee comments**:
L252    2.2.7 should be 2.2.5.2
**Author's reply and actions:**   We thank the referee, We corrected all the sub sections' numbers.

**ID: R1_20**
**Referee comments**:
L268. Which GIS algorithm?
**Author's reply  and actions:**  We added some lines (384-388) to better explain the GIS algorithm that allowed to create the perturbed generic i-DTM of the ensemble generating a vertical error with a normal distribution characterized by a zero mean and a variance that is uniformly distributed with a specific CDE: "*The above-mentioned GIS algorithm includes the following steps: 1. Generation of a raster (NR) of random values with a normal Gaussian distribution ($\mu$=0, s=1) for the entire extension of the DTM; 2. generation of a raster(SR) with the average of the NR values within a neighborhood equal to CDE; 3. creation of an error distribution raster (Err) dividing the SR raster by its spatially averaged standard deviation and multiplying the result for the adopted variance (U(0,0.3));4. addition of the Err raster to the original DTM*"

**ID: R1_21**
**Referee comments**:
L288 agricoltural -> agricultural
**Author's reply and actions:**  Thank you, corrected. (line 407)

**ID: R1_22**
**Referee comments**:
Figure 5: An issue with this plot is that from the manuscript it is unclear how often are the observations, and as this seems to be both an "observed" location whose information is assimilated into the model and a location with observations against which the model forecast is evaluated, it is difficult (if possible) to disentangle the forecast skill here, or how much of the assimilated run line comes from the gauge observations themselves.
**Author's reply and actions:**  We thank the Referee for the comment, we agree that the model performances should be evaluated also far from the assimilation location. We produced a new set of results  where we assimilated only the upstream stage  gages observations (ASS 1SG) to evaluate how the performances behave and vary downstream. We kept also the results on the assimilation of the 4 stage gages (ASS 4SG) to show how the covariance and the model uncertainty behave along the computational domain. The frequency of the assimilate observation is 15 minutes (very frequent as respect to the plots scale), therefore we deicide to not show the frequency on the plot, but we specified the frequency in the caption.

**ID: R1_23**
**Referee comments**:
Figure 8 & Section 4.2: The lack of persistence in the benefit in assimilation the satellite observation seems directly related to the white noise mentioned above. Consider time autocorrelated inflow errors.
**Authors' reply:** We thank the Referee for the comment. According to the Referee's suggestion we added the time autocorrelated inflow errors. However, the lack of persistence is consistent with other outcomes in scientific literature where hydraulic model states are updated (few hours or even minutes, e.g. Andreadis et al., 2007; Matgen et al., 2010; García-Pintado et al., 2013; Andreadis and Schumann, 2014). Some of these studies demonstrated that the updating of inflows boundaries (with and augmented equation)  can increase the persistence of the errors reductions between the

observations in both 1D and 2D models. This can be a future improvement of the proposed work, as specified in Section 4.4

**2) Referee #2**

**ID: R2_01**
**Referee comments**:
This study aims at simultaneously assimilating water level observations from static sensors and EO-derived flood extent for improving real-time flood modeling. I have really enjoyed reading this paper, which deals with a timely and important issue. The authors showed the potential of the joint assimilation of water level observations from both static sensors and satellite images. I think this study fits the overall focus of HESS. However, I do have a number of major comments that hopefully will help the authors in strengthening their manuscript.
**Authors' reply:** We thank the reviewer Dr. Maurizio Mazzoleni for the positive feedbacks and the useful suggested revisions that helped improving the manuscript
**Actions:** We isolated every Referee comment assigning a specific ID with a progressive number (e.g. R1_XX) and our point-by-point reply.

**ID: R2_02**
**Referee comments**:
-My first comment concerns the overall objective of this study. Personally, I would put more emphasis on the issue of the joint assimilation of water level observations in the 1-D and 2-D model rather than highlighting the innovation behind the proposed DA approach (line 57). Besides equations 5 and 6, and the definition of hko,t in equation 9 there is no much difference between a standard EnKF and the proposed DA method, which would not justify a publication on a high impact journal like HESS. To the best of my knowledge, this is the first study that assimilates heterogeneous observations in both 1D and 2D models and this must be better highlighted in the introduction (as novelty and the main objective of the paper) and throughout the paper.
**Authors' reply:** We agree with the Referee,'s comment:  from the introduction, The EnKF is, in fact, a consolidated methodology, therefore we underlined  the joint assimilation of stage gauges and satellite derived flood extents adopting novel methodologies for updating the Quasi-2D hydraulic model.
**Actions:** We removed the "novel" word in line 111 and we better clarified the novel aspects of the proposed research (lines 109-130)*: "'Despite the remarkable progress in the integration of remotely sensed observations in DA frameworks, there are still major challenges that need to be faced (Grimaldi et al., 2016). For example, there is not still in scientific literature an approach able to assimilate heterogeneous observations from both local and distributed datasets coming from different sources (i.e. traditional stage gauges and remotely sensed flood extents). Moreover, Quasi-2D and 2D hydraulic models can be sensitive to different simultaneous local state updating (i.e. water level corrections at specific time steps), because contiguous channel/floodplain cells can be characterized by different elevations, geometry and roughness, therefore instability issues can rise during the model corrections. Another critical issue is that large scale flood forecasting models need to  provide timely predictions but  their spatial resolution can limit the effectiveness of the assimilation of satellite derived flood extents (Hostache et al., 2018).*
*In this work, a DA framework supported by heterogeneous observations coming from both local water level observations (i.e. stage gauges) and spatially distributed information gathered from satellite images - is proposed and tested. This research seeks to develop a more flexible DA scheme that may value all available sources of observations for distributed flood modelling updates. The aim of this work is to mitigate flood prediction uncertainties by combining heterogeneous data and an integrated topographic-hydrologic-hydraulic modelling approach, while maintaining inundation forecasting robustness, scalability and numerical stability. In achieving this goal, novel scientific advances and technical challenges of EO-driven DA approaches for flood prediction are investigated and in particular: A methodology for updating the state*

*variable from multiple local stage gauges observations of a hydraulic model for distributed flood routing in floodplain domains; the gathering of spatially distributed water level observations by means of flood extension processing and detection from satellite images, also adopting GIS algorithms for overcoming the issues of the different resolutions between the ensembles of the flood extents retrieved from the satellite derived images and the ones generated from the hydraulic model simulations.* "
"

**ID: R2_03**
**Referee comments**:
- The proposed DA approach should be better described in the paper. What I think is still missing is the information about the size of each DA variable/matrix (e.g. the size of the model covariance matrix P) and how the merging between hydraulic model and DA is performed. Observations from static sensors are used to update the channel water level (1-D model), while satellite images are used for updating the floodplain water level (2-D model). The assimilation of one observation at a given time step allows updating not only the water level at that specific point along the channel but also upstream and downstream. This is partially solved by introducing the distributed gain (initially proposed in Madsen and Skotner, 2005), but how then the updated upstream flow will numerically influence the downstream water levels? It would be nice to show the covariance matrix P at different time steps in case of assimilation of only static sensor, only SI, and joint assimilation. This will allow visualizing the distributed effect of assimilating heterogenous observations at once.

**Authors' reply and actions:** We thank the referee for the useful comments. We extended Section 2.2.1.1 for better explaining how the model updating is performed at the assimilation steps. The model updating are applied "serially", allowing to reduce the DA variable matrix to sequences of one observation at time and avoiding potential spurious correlations of observations located far from each other. This serial updating is commonly used also in observation localization techniques where, for example, at each point of the domain, the covariance of the observation is divided by a term that is inversely proportional to the inverse of a distance-based correlation. We also clarified that in both cases of assimilating satellite derived images or stage gauges observation, the model updating is performed in both channel and floodplain cells. We also set a new simulation in which only the upstream SG observations are observed in order to show the performance in the downstream part of the basin, far from the observation locations. Finally, we also add 2 new figures to show the distribution of the covariance matrix at specific time step

**ID: R2_04**
**Referee comments**:
- The abstracts read well but I would include a couple of brief sentences summarizing (quantitatively) the benefits of the joint assimilation (e.g. "Our findings reveal that assimilating observations from static sensors and satellite led to an overall reduction of the Bias and RMSE of about ---" ). In addition, at the beginning and at the end of the abstract you referred to the issue of data scarcity. However, your approach is based on the case in which you have observations from static sensors, which may be not available in data-scarce regions.

**Authors' reply and actions:** We added some lines in the abstract specifying some quantitative findings of the proposed approach (lines 18-21). We mentioned the issue of data scarcity because our proposed methodology is able to work even if gauging stations are missing and satellite derived data are the only sources of observations.

**ID: R2_05**
**Referee comments**:

- In line 143 the authors state that "In case the observation is a stage gauge measurement, the state variable position is determined by identifying the closest channel cell". However, after a few lines (153) they stated "The updating of the water levels from Static Sensors (SH) [...] aims to correct both the channel and the floodplain water level". Are the static water level observations used to update only channel water levels of also the ones in the floodplains?

**Authors' reply:** We better clarified this aspect in Section 2.2.1.1 (see also in Figure 2): the model updating when stage gauges observations are assimilated is performed both in the Channel and in the closest floodplain cells and is propagated upstream and downstream in both channel and floodplain. This helps preventing model instabilities if only channel cells are updated and not the adjacent floodplain cells.

**Actions:** We better specified this aspect referring to Section 2.2.1.1. (lines 199-201): *"The correction is then applied also to the closest floodplain cells and propagated upstream and downstream as illustrated in Section 2.2.1.1 ."*

**ID: R2_06**
**Referee comments**:
- I like the way the different experiments are structured and described. However, I think that a more critical analysis of the results is needed. I would like to see more discussion on results achieved with the assimilation of SI observations. The description of the results is there but what is lacking is the "why" you got these results. For example, assimilating SG observations we see that the ensemble with DA is similar to the one of OL in the downstream area of Figure 6 (see lines in lines 385-388). However, this is not the case when assimilating SI observations (figure 9). Figure 6 is barely discussed in the paper, so figure 9. Including the spatial values of P and K may help in understanding this behavior and better describe the results.

**Authors' reply and actions:** We thank the Referee for the suggestion. We extended the description and the discussion on the results (lines 490-585) and we also added three Figures representing the spatial covariances at specific time steps.

**ID: R2_07**
**Referee comments**:
- Could you elaborate more on the impact of the low retrieval frequency of SI observations on the DA performances?

**Authors' reply:** Since our case study included the acquisition of only one satellite image, we could not analyze the impact on the low frequency acquisition of the SI observation. However, recent scientific literature provided some important findings on the frequency of the SI acquisition. For example, Dasgupta et al., 2021 found that the optimal strategy for the image acquisition depends on the river morphology and flood wave arrival timing. Moreover, it was found that the number of observations to significantly improve the performances of the DA model increase with the narrowing of the floodplain valley. Moreover, Giustarini et al., 2011 found that the frequency of their model corrections seems to be effective mostly during the rising limb of the flow hydrograph, while it seemed not to be significantly efficient during the recession limb.

**Actions:** We specified these aspects in lines 74-77 and in lines 104-108.

**ID: R2_08**
**Referee comments**:
- Is your DA approach efficient when dealing with high-dimensionality issues of the covariance matrix P?

**Authors' reply:** As specified in comment R2_03, the application of the DA model is performed applied "serially" for each observation, allowing to reduce the DA variable matrix to sequences of one

observation and avoiding potential spurious correlations of observations located far from each other. Therefore, there are not high-dimensionality issues of the covariance matrix.
**Actions:** We specified this aspect in Section 2.2.1.1 (lines 239.242 and 266-268)

**ID: R2_09**
**Referee comments**:
- What is the computational time required to run the DA approach in the selected case study?
**Authors' reply:** Averagely, each simulation hour require a computational time equal to 3.7 minutes. This is a value averaged considering that the computational time is highly variable depending on the peak flow and on the extension of the flooded area in the computational domain
**Actions:** This information is added in the manuscript in Section 4.4 (lines 613-618)

**ID: R2_10**
**Referee comments**:
- What is the difference between SH and SG? Try to avoid unnecessary acronyms if not used.
**Authors' reply:** We apologize for the typos
**Actions:** SH has been replaced with SG

**ID: R2_11**
**Referee comments**:
- Why do you get such an abrupt change in Figure 13?
**Authors' reply:** This abrupt change was in correspondence of the SI acquisition that determine an abrupt reduction of the ensemble spread
**Actions:** We changed the way of simulating the simultaneous assimilation of SG and SI observation without assuming SG failures, therefore new different results are showed.

**ID: R2_12**
**Referee comments:**
- Where is the text of sections 2.2.1 and 2.2.5?
**Authors' reply:** There was a mistake in the subsections labelling.
**Actions:** subsections' names have been corrected

**ID: R2_13**
**Referee comments**:
- Why are the results of OL in tables 1, 2, and 3 different? I would expect the same values if the sensor location and flood events are the same.
**Authors' reply:** The slight differences between the OL results in the three tables were due to the fact that we repeated three different sets of the 2021 flood event simulations even for the OL. Since the OL each time is characterized by the generation of the model and observation errors, slight difference may occur, mostly if the sample size is limited for computational reason.
**Actions:** We referred to the same OL simulation so as not to confuse the reader

**ID: R2_14**
**Referee comments**:
- Line 388: "The adopted updating procedure allows to increase the flood extent of 4 km2 a the time of the SI acquisition". Is this increment leading to better prediction or more false alarms?
**Authors' reply:** The updating procedure helped reducing the false negatives
**Actions:** We specified this aspect in Section 4.2 (lines 545-548)